# A parameterized two-domain thermodynamic model explains diverse mutational effects on protein allostery

Zhuang Liu[1], Thomas G Gillis[2], Srivatsan Raman[2,3,4], Qiang Cui[1,5]*

[1]Department of Physics, Boston University, Boston, United States; [2]Department of Biochemistry, University of Wisconsin, Madison, United States; [3]Department of Chemistry, University of Wisconsin, Madison, United States; [4]Department of Bacteriology, University of Wisconsin, Madison, United States; [5]Department of Chemistry, Boston University, Boston, United States

**Abstract** New experimental findings continue to challenge our understanding of protein allostery. Recent deep mutational scanning study showed that allosteric hotspots in the tetracycline repressor (TetR) and its homologous transcriptional factors are broadly distributed rather than spanning well-defined structural pathways as often assumed. Moreover, hotspot mutation-induced allostery loss was rescued by distributed additional mutations in a degenerate fashion. Here, we develop a two-domain thermodynamic model for TetR, which readily rationalizes these intriguing observations. The model accurately captures the in vivo activities of various mutants with changes in physically transparent parameters, allowing the data-based quantification of mutational effects using statistical inference. Our analysis reveals the intrinsic connection of intra- and inter-domain properties for allosteric regulation and illustrate epistatic interactions that are consistent with structural features of the protein. The insights gained from this study into the nature of two-domain allostery are expected to have broader implications for other multi-domain allosteric proteins.

## eLife assessment

The study presents **valuable** findings where two-domain thermodynamic model for TetR accurately predicts in vivo phenotype changes brought about as a result of various mutations. The evidence provided is **compelling** and features the first innovative observations with a computational model that captures the structural behavior, much more than the current single-domain models.

## Introduction

Allostery, a fundamental regulatory mechanism of biomolecular functions, is prevalent in life processes (*Monod et al., 1965*; *Koshland et al., 1966*; *Changeux and Edelstein, 2005*; *Cui and Karplus, 2008*; *Motlagh et al., 2014*; *Yu and Koshland, 2001*; *Süel et al., 2003*; *Dokholyan, 2016*). The long-range signaling of allostery makes it a fascinating phenomenon, in which binding of an effector molecule (ligand) at the allosteric site alters the function of a distal active site (*Dokholyan, 2016*; *Leander et al., 2020*; *Peracchi and Mozzarelli, 2011*; *Wodak et al., 2019*). Current descriptions of allostery can be largely cast into two categories: one adopts a mechanical view, focusing on the propagation of conformational distortions from the allosteric site to the active site (*Wang et al., 2020*; *Lockless and Ranganathan, 1999*; *Daily and Gray, 2009*; *Rodriguez et al., 2010*; *Lee et al., 2008*; *Walker et al., 2020*); the other emphasizes the thermodynamic aspect of the problem, highlighting the effect of ligand binding on shifting the protein population among pre-existing conformational

*For correspondence:
qiangcui@bu.edu

states characterized by different ligand binding affinities and active site properties (e.g. the classic MWC model) (*Monod et al., 1965*; *Koshland et al., 1966*; *Changeux and Edelstein, 2005*; *Cui and Karplus, 2008*; *Sevvana et al., 2012*; *Takeuchi et al., 2019*; *Marzen et al., 2013*). Models based on both perspectives have provided insights into the function of prototypical allosteric systems, thanks to decades of combined efforts of experiment, computation, and theory (*Cui and Karplus, 2008*; *Motlagh et al., 2014*; *Dokholyan, 2016*; *Marzen et al., 2013*; *Changeux, 2012*; *Guo and Zhou, 2016*; *Schueler-Furman and Wodak, 2016*; *Reichheld et al., 2009*; *Xu et al., 2003*; *Nierzwicki et al., 2021*; *East et al., 2020*). Of note, these two perspectives of allostery are complementary rather than contradictory to each other (*Liu and Nussinov, 2016*). The conformational coupling between spatially distant functional sites (allosteric site and active site) plays a vital role in regulating allosteric function, allowing for the transmission of signals from one site to the other (*Zhang et al., 2020*). While the mechanical view primarily seeks to identify the structural basis for signal transduction, it is implicitly assumed within the population shift perspective, which offers a comprehensive and quantitative description of allostery (*Szabo and Karplus, 1972*; *Viappiani et al., 2014*; *Henry et al., 2020*; *Eaton, 2022*).

In recent years, a thermodynamic model referred to as the ensemble allosteric model (EAM) has been applied to conceptualize protein allostery in terms of intra- and inter-domain properties, with the latter explicitly quantifying the energetic coupling between distant functional sites (*Motlagh et al., 2014*; *Wodak et al., 2019*; *Hilser et al., 2012*). This framework is consistent with the observation that allosteric proteins often partition different activities into distinct domains, such as the ligand- (LBD) and DNA-binding domains (DBD) in transcription factors (TFs) and the effector-binding and catalytic domains in enzymes (*Ramos et al., 2005*; *Tzeng and Kalodimos, 2012*; *Velyvis et al., 2007*; *Lipscomb and Kantrowitz, 2012*). Such thermodynamic approach finds broad applicability across proteins in general, as it has been proposed that all proteins are potentially allosteric (*Zhang et al., 2020*; *Gunasekaran, 2004*). Consequently, this raises intriguing questions about the nature of allostery. For instance, do intrinsic connections exist between the intra- and inter-domain properties of a protein, given the highly cooperative nature of allosteric networks? What roles do sequence and structure play in synergistically determining these properties? Furthermore, what are the parameters within the model that are most essential to the accurate description of realistic allosteric systems, especially prediction of activity upon multiple mutations? To answer these questions and deepen our understanding of allosteric regulation, it is essential to parameterize and test the thermodynamic model using comprehensive mutational data, a topic that still requires further exploration (*Leander et al., 2022*; *Leander et al., 2020*).

A critical test of any thermodynamic model is whether it can explain the effect of mutations on allosteric signaling. Deep mutational scanning (DMS) analysis has emerged as a powerful function-centric approach over the past decade, which measures the impact of all possible single mutations (*Fowler et al., 2010*; *Fowler and Fields, 2014*; *Sarkisyan et al., 2016*; *Flynn et al., 2020*; *Starr et al., 2020*; *Huss et al., 2021*). The methodology provides an unbiased way of identifying critical residues for protein allostery and generates extensive data for validating existing computational and theoretical models. Along this line, recent DMS study of four homologous bacterial TFs in the TetR family (TetR, TtgR, MphR, and RolR) revealed that the residues critical for allosteric signaling (hotspots) in these TFs are broadly distributed with no apparent structural link to either the allosteric or the active site (*Leander et al., 2022*; *Tack et al., 2021*; *Faure et al., 2022*; *Jones et al., 2020*; *McCormick et al., 2021*). This contrasts the commonly held view that hotspot residues tend to form well-defined pathways linking the two sites (*Süel et al., 2003*; *Ota and Agard, 2005*; *Strickland et al., 2008*; *Reynolds et al., 2011*; *Amor et al., 2016*); the observations also hinted at common molecular rules of allostery in the TetR family of TFs (*Cuthbertson and Nodwell, 2013*; *Fukami-Kobayashi, 2003*). Moreover, systematic analysis of higher order TetR mutants in the background of five noninducible ('dead') mutants revealed remarkable functional plasticity in allosteric regulation (*Leander et al., 2020*). Specifically, the loss of inducibility due to the mutations of allostery hotspots could be rescued (restored wildtype [WT]-like inducibility) by additional mutations, and different dead mutations exhibit varying numbers of rescuing mutations, which are usually distal and lacking any obvious structural rationale.

While the identification of broadly distributed dead and rescuing mutations in these studies is exciting, such qualitative activity characterization of mutants (inducible and noninducible at a given

ligand concentration) prevents a deeper mechanistic understanding of the observation. In fact, mutations exert a graded impact on the allosteric signaling of the TFs; i.e., the expression level of the regulated gene varies among both inducible and noninducible mutants in a continuous, ligand concentration-dependent manner. The nuanced mutational effect on allosteric regulation is linked apparently to the equilibria among different conformational and binding states of the mutant TF, which are determined by the allosteric parameters (intra- and inter-domain properties). Therefore, mapping the mutants onto the parameter space of a biophysical model by exploiting additional experimental data (see below) is crucial for elucidating the observed allosteric phenomena in a comprehensive and physically transparent manner.

In this study, we develop a two-domain statistical thermodynamic model for TetR, in which the protein is generically divided into LBD and DBD. Our model incorporates three essential biophysical parameters that capture the intra- and inter-domain properties of the protein, as elaborated in the subsequent section. The model readily rationalizes the myriad ways that mutations can perturb allostery as observed in the aforementioned DMS measurements, by revealing that mutations perturbing intra- and inter-domain properties can lead to similar TF inducibilities at a single ligand concentration. Moreover, the model elucidates the distinct influences of these parameters on the complete induction curve, serving as a diagnostic tool for dissecting the intricate allosteric effects of mutations. We validate the model by accurately describing the induction curves of a comprehensive set of TetR mutants, thereby enabling the quantification of mutational effects and epistatic interactions (*Daber et al., 2011*; *Chure et al., 2019*). The insights from this combined theoretical and

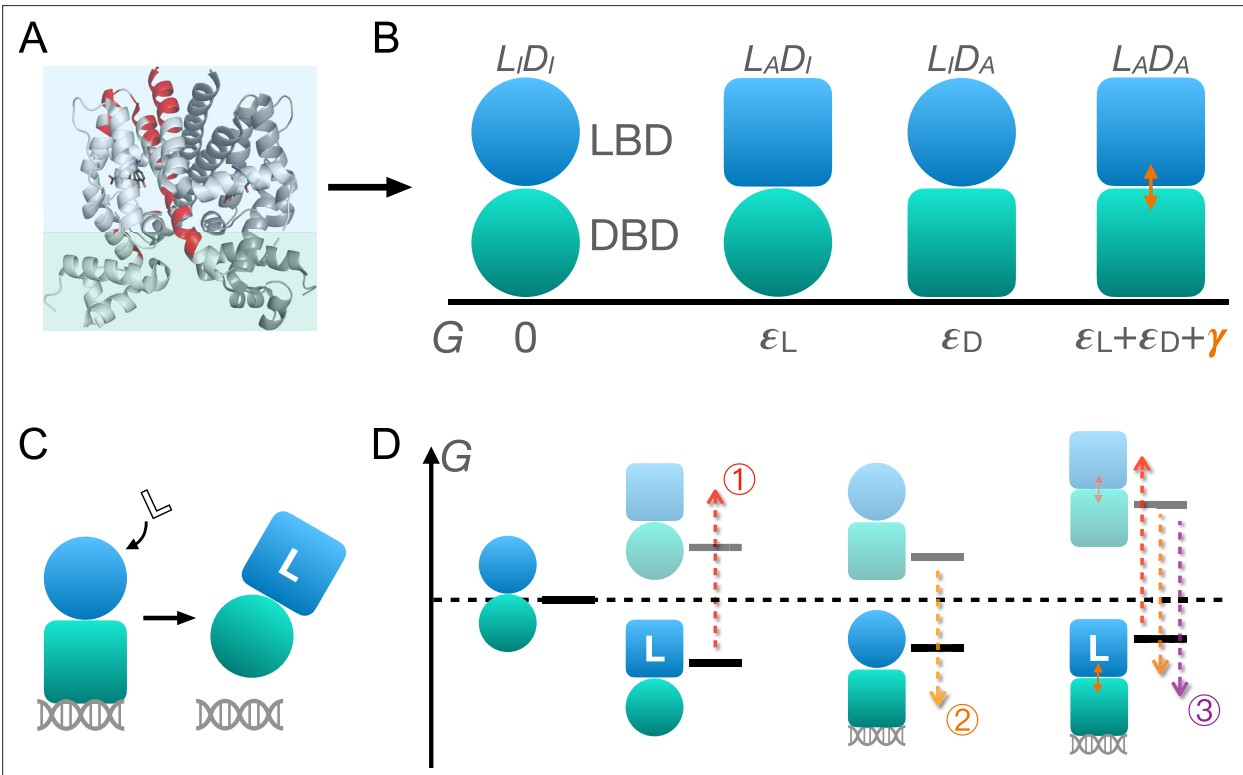

**Figure 1.** Schematic illustration of the two-domain statistical thermodynamic model of TetR allostery. (**A**) The crystal structure of TetR(B) in complex with minocycline and magnesium (PDB code: 4AC0). The red residues are the hotspots identified in the deep mutational scanning (DMS) study (*Leander et al., 2020*). (**B**) Four possible conformations of a two-domain TetR molecule with their corresponding free energies ($G$). $G$ of the $L_ID_I$ state is set to 0. Blue/green circle (square) denotes the inactive (active) state of ligand/DNA-binding domain (LBD/DBD). (**C**) A simple repression scheme of TetR function. Binding of the ligand (inducer) favors the inactive state of DBD in TetR, which then releases the DNA operator and enables the transcription of the downstream gene. (**D**) Schematic free energy diagram of the possible binding states of TetR at fixed ligand and operator concentrations. Red, orange, and purple arrows show how a mutation can disrupt allostery by (1) increasing $\varepsilon_L$; (2) decreasing $\varepsilon_D$, and (3) decreasing $\gamma$. The $L$-$L_AD_A$ state is not explicitly shown in the last column as the doubly bound $L$-$L_AD_A$-$D$ state is expected to have a lower free energy. Note that mutations that change the binding affinities of the active LBD/DBD to ligand/operator are not discussed here as we focus on the intrinsic allosteric properties of the transcription factor (TF) itself.

experimental investigation of TetR allostery are expected to generally apply to other two-domain allosteric systems, such as TFs like catabolite activator protein (CAP) (*Tzeng and Kalodimos, 2012*), receptors like pentameric ligand-gated ion channels (pLGICs) (*Sauguet et al., 2015*; *Hu et al., 2020*), and allosteric enzymes like aspartate transcarbamoylase (ATCase) (*Lipscomb and Kantrowitz, 2012*; *Velyvis et al., 2007*).

## Results

### Overview of the two-domain thermodynamic model of allostery

As shown in *Figure 1A and B*, TetR can be generically divided into a LBD and a DBD, disregarding its homodimeric nature (*Leander et al., 2020*; *Takeuchi et al., 2019*; *Reichheld et al., 2009*; *Leander et al., 2022*; *Yuan et al., 2022*; *Scholz et al., 2004*). In the same vein of the classic allostery models (*Monod et al., 1965*; *Koshland et al., 1966*), each domain features two (the relaxed/inactive and tense/active) conformations that differ in free energies and binding affinities. For the simplicity and mechanistic clarity of the model, we assume negligible binding affinity of the LBD to the ligand and the DBD to DNA in their inactive conformations and consider competent binding only for the active ones. Each domain must overcome a free energy increase to transition from the inactive to the active conformation ($\varepsilon_L$ for LBD and $\varepsilon_D$ for DBD). Importantly, the two domains are allosterically coupled, in that there is a free energy penalty $\gamma$ when both domains adopt the active conformations simultaneously. For example, when the ligand binds to the LBD, selecting its active conformation, it discourages the active conformation of the DBD and therefore DNA binding. This anti-cooperativity establishes the foundation for allosteric regulation in TetR.

Accordingly, there are four possible conformational states of TetR, namely $L_I D_I$, $L_A D_I$, $L_I D_A$, and $L_A D_A$, with $L/D$ and $I/A$ denoting LBD/DBD and inactive/active conformation, respectively (see *Figure 1B* and the upper four states of *Figure 1D*). Binding of ligand/operator to the active LBD/DBD further lowers the free energy of the corresponding state in a concentration-dependent manner following the standard formulation of binding equilibrium (see the lower three states of *Figure 1D*). The regulatory mechanism of TetR allostery can then be qualitatively explained by the schematics in *Figure 1C and D*. Without the ligand, WT repressor predominantly binds to the operator sequence, which obstructs the binding of RNA polymerase (RNAP) to the adjacent promoter of the regulated gene (*Figure 2—figure supplement 1*). In the presence of ligand (inducer) at a sufficiently high concentration, the ligand-bound $L_A D_I$ state ($L\text{-}L_A D_I$) has the lowest free energy compared with other possible (DNA-bound) states, thus the repressor predominantly releases the operator upon ligand binding, enabling the expression of downstream genes (*Figure 1C* and *Figure 2—figure supplement 1*).

Mutations perturb the three biophysical parameters ($\varepsilon_D$, $\varepsilon_L$, and $\gamma$) and hence the free energy landscape (*Figure 1D*), leading to changes in TetR function. The three limiting scenarios are depicted in *Figure 1D*: (1) when a mutation significantly increases $\varepsilon_L$, it raises the free energy of the $L\text{-}L_A D_I$ state relative to the DNA-bound state $L_I D_A\text{-}D$, which results in a dead (noninducible) phenotype; (2) when a mutation substantially decreases $\varepsilon_D$, it raises the free energy of the $L\text{-}L_A D_I$ state relative to both DNA-bound states ($L_I D_A\text{-}D$ and $L\text{-}L_A D_A\text{-}D$), also leading to a dead mutant; (3) mutations that decrease $\gamma$ greatly lower the free energy of the double-bound state ($L\text{-}L_A D_A\text{-}D$) relative to $L\text{-}L_A D_I$, again leading to the suppression of induction. Therefore, these schemes highlight that hotspot mutations can disrupt allostery by perturbing either intra- ($\varepsilon_D$, $\varepsilon_L$) or inter-domain ($\gamma$) properties. Functional readouts like DMS can help distinguish these mechanistic differences at the biophysical level. Notably, mutations mainly perturbing $\varepsilon_D$ or $\varepsilon_L$ do not have to lie on the structural pathways linking the allosteric and active sites, which explains the broad hotspot distributions observed in the DMS measurement (*Leander et al., 2020*; *Reichheld et al., 2009*; *Leander et al., 2022*; *Scholz et al., 2004*). In general, a dead mutation is likely of mixed nature, as long as its effect on the free energy landscape promotes the dominance of the DNA-bound states. Likewise, rescuing mutations may restore WT-like inducibility by modifying intra- and inter-domain energetics in various ways, as far as the dominance of the $L\text{-}L_A D_I$ state is re-established, rationalizing the broad rescuing mutation distributions. Naturally, the rescuability of a dead mutation depends on which biophysical parameters it perturbs and by how much. At a qualitative level, the distributed nature of dead and rescuing mutations observed in the DMS measurement emerges naturally from the two-domain model (*Leander et al., 2020*).

In summary, the interplay of intra- and inter-domain properties that govern the free energy landscape of TetR's conformational and binding states, as elucidated by the two-domain model, aligns well with the unexpected DMS results on a qualitative level. To gain deeper insight into TetR allostery and the model itself, we aim to establish a quantitative framework with the model that accurately captures mutant induction curves in the next section.

## System-level ramifications of the two-domain model

The discussions in the previous section and *Figure 1* are primarily meant to give an intuitive and qualitative understanding of the two-domain model and rationalization of recent DMS measurements (*Leander et al., 2020*; *Leander et al., 2022*). In this section, we further establish a quantitative connection between the model and the induction curves of TetR variants, leveraging the recent success of linking sequence-level perturbations to system-level responses (*Daber et al., 2011*; *Chure et al., 2019*; *Garcia and Phillips, 2011*; *Brewster et al., 2014*; *Weinert et al., 2014*; *Rydenfelt et al., 2014*; *Razo-Mejia et al., 2017*). An induction curve describes the in vivo expression level of the TetR-regulated gene as a function of inducer concentration.

As described in the last section, the intra-domain parameters ($\varepsilon_D$, $\varepsilon_L$) and inter-domain parameter ($\gamma$) of the two-domain model collectively determine the free energy landscape of TetR (*Figure 1*). Consequently, a degenerate relationship arises between combinations of parameter values and the induction level at a specific ligand concentration as measured in the DMS study. To tease apart the allosteric effects of different parameters, we aim to formulate their connection to the full induction curve, which is characterized by (1) the expression level of the TF-regulated gene without ligand (leakiness), (2) gene expression level at saturating ligand concentration (saturation), and (3) the ligand concentration required for half-maximal expression ($EC_{50}$). As revealed in previous studies, these induction curve properties encode information for the key parameters of an allosteric regulation system, e.g., the TF-binding affinity to ligand and operator, equilibrium between different conformational states of the TF and the abundance of various essential molecules within the system. The establishment of a quantitative connection between the MWC model and the induction curve has yielded valuable insights into several allosteric systems (*Eaton, 2022*; *Daber et al., 2011*; *Chure et al., 2019*). However, the analogous aspects within a multi-domain thermodynamic model for allostery require further investigation.

Inspired by the pioneering works on transcription with MWC models (*Daber et al., 2011*; *Chure et al., 2019*), we derived a quantitative relation between the gene expression level in cells and the three biophysical parameters of the two-domain model (*Equation 1*). Briefly, the ratio of the expression level of a TF-regulated gene to that of an unregulated gene, termed fold change ($FC$) and bound between 0 and 1, is used to quantify gene expression. The gene expression level on the other hand is assumed to be proportional to the probability that a promoter in the system is bound by RNA polymerase, p(RNAP). Thus, $FC$ is evaluated as the ratio of p(RNAP) in the presence of the TF to that in the absence of the TF. Intuitively, p(RNAP) can be calculated based on the equilibria among different binding and conformational states of the TF (*Figure 2—figure supplement 1* and *Figure 2—figure supplement 2*), which are determined by the allosteric properties ($\varepsilon_D$, $\varepsilon_L$, and $\gamma$) and several other parameters. Therefore, $FC$ can be expressed as a function of these parameters, which, under the assumption that intra- and inter-domain energetics adopt typical values of several $k_B T$ , is simplified to *Equation 1*.

$$FC = \left( 1 + R^* e^{-\varepsilon_D} \frac{1 + e^{-\varepsilon_L - \gamma}(\frac{c}{K})^2}{1 + e^{-\varepsilon_L}(\frac{c}{K})^2} \right)^{-1} \tag{1}$$

Here, $R^*$ is the rescaled TF copy number in the cell; $c$ is ligand concentration, and $K$ is the dissociation constant of ligand to the repressor with an active LBD. As we focus on understanding how sequence-level perturbations are manifested in the allosteric properties of the protein (depicted by $\varepsilon_D$, $\varepsilon_L$, and $\gamma$), the residues we choose for mutational study here are mostly not in direct contact with either the ligand or the operator (see Appendix 1 for more discussions). Thus, $K$ and $R^*$, which is determined by the TF copy number in the cell and the affinity of the operator to the repressor with an active DBD, are taken to be constants across mutations in all subsequent discussions (*Chure et al., 2019*). Detailed derivations of *Equation 1* are provided in Appendix 1.

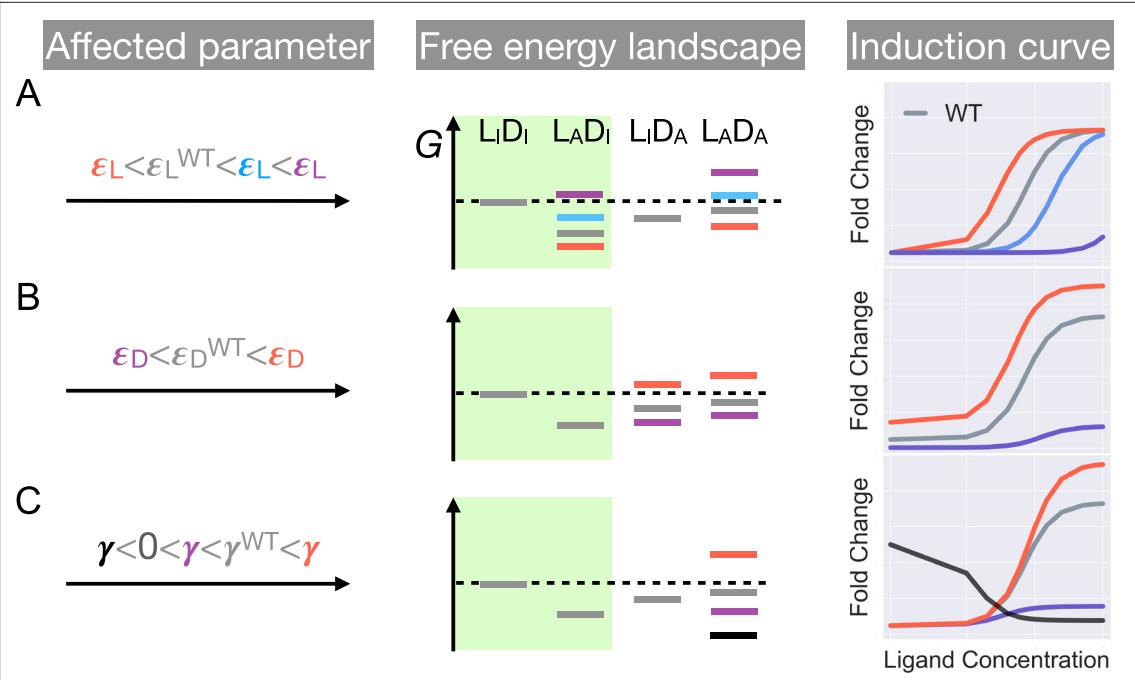

**Figure 2.** Schematic illustration of the characteristic effects of perturbations in the three biophysical parameters on the free energy landscape and the corresponding induction curves of a two-domain allosteric system. Panels A, B, and C illustrate how changing $\varepsilon_L$, $\varepsilon_D$, and $\gamma$ alone affects the free energy landscape for the binding states shown in *Figure 1D* and the induction curve. For the black induction curve in (C), the values of $\varepsilon_L$ and $\varepsilon_D$ are also adjusted to aid visualization of the negative monotonicity of the gene expression level (fold change) as a function of ligand concentration. The green shade in the middle column separates the DNA-bound states from the rest. In the free energy landscapes shown in the middle column, ligand or DNA binding is always assumed when the corresponding domain is in the active conformation.

The online version of this article includes the following figure supplement(s) for figure 2:

**Figure supplement 1.** Statistical weights of promoter occupancy states and repressor states.

**Figure supplement 2.** Equilibria among different conformational and binding states of the repressor.

**Figure supplement 3.** Extended parametric study of main text *Equation 1*.

Despite the considerable degeneracy in the activity (inducible and noninducible) within the parameter space of the two-domain model (*Figure 1D*), *Equation 1* demonstrates that mutations affecting distinct biophysical parameters can be discerned based on their characteristic effects on the induction curve.

First, as $\varepsilon_L$ and $\gamma$ have no effect on the leakiness of the induction curve, only mutations that modify $\varepsilon_D$ can lead to its changes (*Figure 2* and Appendix 1 for additional discussions). In addition, these mutations also change the level of saturation ($FC$ value where the induction curve plateaus at large $c$). Thus, dead mutations that disrupt allostery by decreasing $\varepsilon_D$ alone will uniquely feature a noticeably lower leakiness and a lower level of saturation compared with the WT (*Figure 1D* and *Figure 2*, see also Appendix 1 and *Figure 2—figure supplement 3* for more discussions). Second, mutations that solely perturb the other intra-domain property $\varepsilon_L$, a crucial determinant of TetR's ligand detection limit, primarily shift the $EC_{50}$ of the induction curve (see *Equation 1*). As $\varepsilon_L$ increases/decreases from the WT value, the induction curve is right/left shifted with leakiness and the level of saturation remaining unchanged (see the blue and red curves in *Figure 2A*). However, when $\varepsilon_L$ further increases, the induction curve loses the sigmoidal shape, with its sharply varying tail region being the characteristic of a dead mutation that disrupts allostery mainly by increasing $\varepsilon_L$ (purple curve in *Figure 2A*, see Appendix 1 and *Figure 2—figure supplement 3* for more discussions).

Finally, in the high concentration limit, *Equation 1* converges to a constant value (see *Equation 2*).

$$FC = \left(1 + R^* e^{-\varepsilon_D - \gamma}\right)^{-1} \tag{2}$$

Hence, mutations affecting $\gamma$ alone will tune the saturation of a sigmoidal induction curve, which increases/decreases as $\gamma$ increases/decreases (see red and purple curves in *Figure 2C*). Therefore, in contrast to the two aforementioned scenarios, dead mutations that disrupt allostery through decreasing $\gamma$ alone will feature a full sigmoidal induction curve with a low level of saturation and unchanged leakiness compared with the WT (with $\gamma > 0$, see the purple curve in *Figure 2C*). Furthermore, as $\gamma$ dictates the inter-domain cooperativity, it thereby controls the sign of the monotonicity of *FC* as a function of $c$ (see Appendix 1 for detailed proof). Specifically, when $\gamma > / < 0$, there is negative/positive cooperativity between ligand and operator binding, and *FC* increases/decreases monotonically with $c$ (see *Figure 2C*). When $\gamma = 0$, however, the bindings of ligand and operator become independent of each other, and $c$ no longer affects *FC* (*Figure 2—figure supplement 3*).

The distinctive roles of the three biophysical parameters on the induction curve as stipulated in *Equation 1* could be understood in an intuitive manner as well. First, the value of $\epsilon_D$ controls the intrinsic strength of binding of TetR to the operator, or the intrinsic difficulty for ligand to induce their separation. Therefore, it controls how tightly the downstream gene is regulated by TetR without ligands (reflected in leakiness) and affects the performance limit of ligands (reflected in saturation). Second, the value of $\epsilon_L$ controls how favorable ligand binding is in free energy. When $\epsilon_L$ increases, the binding of ligand at low concentrations become unfavorable, where the ligands cannot effectively bind to TetR to induce its separation from the operator. Therefore, the fold change as a function of ligand concentration only starts to noticeably increase at higher ligand concentrations, resulting in larger $EC_{50}$. Third, as discussed above, $\gamma$ controls the level of anti-cooperativity between the ligand and operator binding of TetR, which is the basis of its allosteric regulation. In other words, $\gamma$ controls how strongly ligand binding is incompatible with operator binding for TetR, hence it controls the performance limit of ligand (reflected in saturation).

Having identified the distinctive impacts of the different allosteric parameters of the two-domain model on the induction curve, as outlined in *Equation 1*, we next apply the model to analyze actual

**Table 1.** Distances to the DNA operator and ligand of the 21 residues under mutational study.

| Residue number | Distance to DNA operator (Å) | Distance to ligand (Å) |
|---|---|---|
| 26 | 7.3 | 24.7 |
| 32 | 12.6 | 30.4 |
| 42 | 8.1 | 25.0 |
| 44 | 9.4 | 21.6 |
| 47 | 7.7 | 21.9 |
| 49 | 11.4 | 17.8 |
| 53 | 17.0 | 12.1 |
| 57 | 22.5 | 7.0 |
| 76 | 45.7 | 17.9 |
| 98 | 19.9 | 15.6 |
| 102 | 18.1 | 14.2 |
| 105 | 24.8 | 7.7 |
| 132 | 39.3 | 16.0 |
| 143 | 29.4 | 16.9 |
| 146 | 25.8 | 17.6 |
| 147 | 26.8 | 16.0 |
| 150 | 23.2 | 19.0 |
| 174 | 34.5 | 14.9 |
| 176 | 38.8 | 19.5 |
| 177 | 35.5 | 17.5 |
| 203 | 55.1 | 28.7 |

experimental induction curves of TetR mutants. The analysis enables us to uncover the underlying biophysical factors contributing to diverse mutational effects. Additionally, it allows us to evaluate the model's validity in capturing TetR allostery by assessing its accuracy in reproducing the experimental data.

## Extensive induction curves fitting of TetR mutants

With the diagnostic tools established, we mapped mutants to the parameter space of the two-domain model through fitting of their induction curves. We choose 15 mutants for analysis in this section, which contain mutations that span different regions in the sequence and structure of TetR (*Figure 3—figure supplement 1* and *Table 1*). Five of the 15 mutants consist of a dead mutation G102D and one of its rescuing mutations (see the third row of *Figure 3A*), while the other 10 contains the WT, 8 single mutants and a double mutant Q32A-E147G (see the first two rows of *Figure 3A* and Appendix 1 section 'Mutation selection for two-domain model analysis' for more discussions). In all cases, fitting of an induction curve is divided into two steps: first, at $c = 0$, *Equation 1* can be rearranged to give

$$\varepsilon D = -ln\left(\frac{leakiness^{-1} - 1}{R^*}\right) \tag{3}$$

which enables the direct calculation of $\varepsilon_D$ from the leakiness of the induction curve; second, $\varepsilon_L$ and $\gamma$ are inferred based on the remaining induction data using the method of Bayesian inference (*Chure et al., 2019*). Briefly, given $\varepsilon_D$ calculated in the first step, we select a set of $\varepsilon_L$ and $\gamma$ values from physical ranges of the parameters, based on the probability of observing the induction curve with the parameter values using Monte Carlo (MC) sampling. The medians of the selected sets of $\varepsilon_L$ and $\gamma$ values, which are known as posterior distributions, are reported as the inferred parameter values, and error bars show the 95% credible regions of the posterior distributions (*Figure 3B* and *Figure 4—figure supplement 2B*). All details are provided in the Materials and methods section, Appendix 1, and *Figure 3—figure supplement 2* to *Figure 3—figure supplement 6*.

As shown in *Figure 3A*, the fitted curves accurately capture the induction data of all the mutants studied here, enabling the quantification of their biophysical parameters with little uncertainty (see *Figure 3B* and *Table 2*). These results support the general applicability of the two-domain model to describing TetR allostery, as these 15 mutants vary significantly in terms of location of mutations and phenotype. They are one to four mutations away from the WT, and the sites of mutations are distributed across LBD, DBD, and the domain interface (*Figure 3—figure supplement 1*). In terms of the phenotype, they fall into different classes as characterized by the $FC$ value at $c = 1000$ nM ($FC^{1000}$), including dead ($FC^{1000} < 0.1$), enhanced induction ($FC^{1000} > FC_{WT}^{1000}$), and neutral activity (the rest).

Besides inspecting the goodness of fit, a closer examination of the induction curves and the Bayesian inference results reveal additional features of the two-domain model of TetR allostery. First, the apparent binding affinity of TetR to the ligand (anhydrotetracycline [aTC]) and operator (*tetO2*) used in our experiments are estimated to be about 17.4 $k_B T$ and 16.4 $k_B T$, respectively, based on the parameter values inferred for the WT (see Appendix 1 for detailed calculations); these agree well with the range of reported experimental values (*Scholz et al., 2000*; *Schubert et al., 2004*; *Kedracka-Krok and Wasylewski, 1999*; *Kamionka et al., 2004*; *Bolintineanu et al., 2014*; *Normanno et al., 2015*). Second, as shown in *Figure 3B*, the perturbations in $\varepsilon_D$ are generally small in magnitude (<1 $k_B T$) among the dead mutants, while substantially larger perturbations (up to     5 $k_B T$) are observed for $\varepsilon_L$ and $\gamma$. The latter observation is in line with the predictions of the model (see the previous section) that two types of qualitatively different induction curves are expected for partially dead mutants (see the first and second rows of *Figure 3A*). Specifically, the induction curve of P176N-I174K-F177S (PIF) loses the sigmoidal shape, with its sharply varying tail region suggesting a significant increase of $\varepsilon_L$. This is confirmed by the parameter fitting result, which shows that the triple mutations of PIF, located in the core region of LBD (Appendix 1 and *Figure 3—figure supplement 1*), lead to the largest increase of $\varepsilon_L$ among all mutants (4.5 $k_B T$ higher than the WT, see *Figure 3B*). The induction curves of the other four dead mutants (R49G, D53H, P105M, and G143M), however, maintain the sigmoidal shape yet with low levels of saturation. As they all exhibit a higher $\varepsilon_D$ (leakiness) than the WT, the low levels of saturation have to result solely from weakened inter-domain couplings (*Figure 3*). In other words, these four dead mutations, located mostly at the domain interface (*Figure 3—figure supplement 1*),

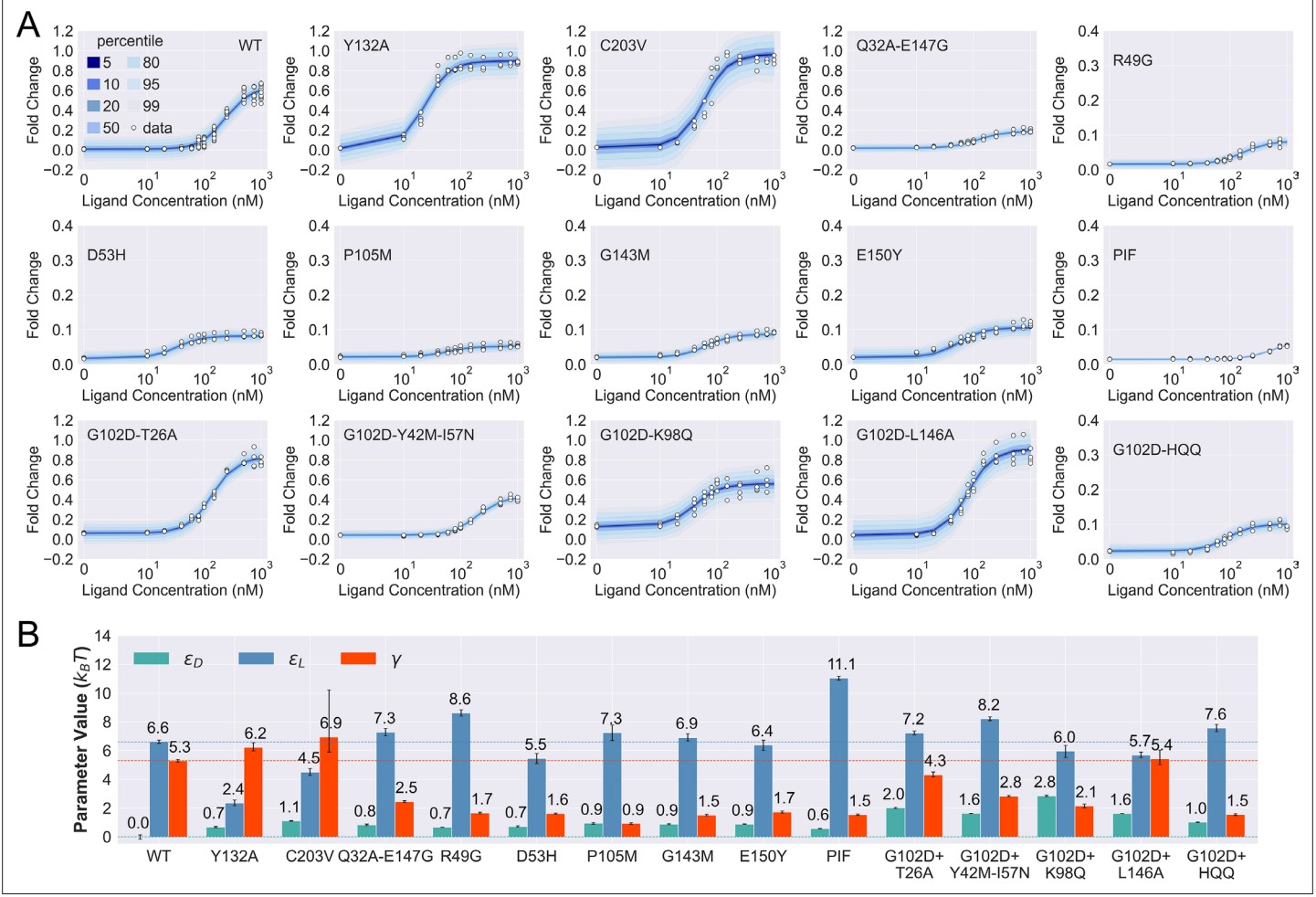

**Figure 3.** Induction data of 15 TetR mutants and the corresponding parameter estimation results. (**A**) Shaded blue curves in each plot show the percentiles of the simulated fold change measurements using the inferred posterior parameters of the mutant. The white data points represent the corresponding experimental induction measurement of four or more biological replicates (three replicates for C203V and G102D-HQQ). (**B**) The inferred parameter values of the 15 mutants. The error bars of $\varepsilon_L$ and $\gamma$ represent the 95th percentile of the Bayesian posterior samples, while the error bar of $\varepsilon_D$ is calculated based on the standard error of the mean (SEM) of the corresponding leakiness measurement. The horizontal lines indicate the wildtype (WT) parameter values for reference.

The online version of this article includes the following figure supplement(s) for figure 3:

**Figure supplement 1.** Sequence and structural distributions of the 21 residues chosen for the mutation analyses in this work.

**Figure supplement 2.** Prior probability distributions and prior predictive check.

**Figure supplement 3.** Probability distributions of $\varepsilon_L$, $\gamma$, and $\sigma$ values in the 1000 sets of prior predictive draws (ground truth) and the average of the corresponding 1000 sets of inferred posterior distributions of the parameters (inferred).

**Figure supplement 4.** Distributions of rank statistics of the prior predictive draws relative to the corresponding posterior samples.

**Figure supplement 5.** Sensitivity analysis for model parameter inference.

**Figure supplement 6.** Posterior predictive check of mutant G102D-Y42M-I57N.

**Figure supplement 7.** Theoretical induction curves of the four dead mutants when their $\gamma$ values are set to the wildtype (WT) value while using their respective $\varepsilon_D$ and $\varepsilon_L$ values (taken from *Figure 3B* in the main text).

**Figure supplement 8.** Sorting scheme to identify dead variants.

disrupt allostery primarily through decreasing $\gamma$. Indeed, when the $\gamma$ of these mutants is set to the WT value but keeping their respective $\varepsilon_D$ and $\varepsilon_L$ parameters unchanged, higher $FC^{1000}$ values than the WT are obtained instead (*Figure 3—figure supplement 7*).

Despite the discussion of two limiting types of changes in the induction curves, it is interesting to observe in *Figure 3B* that the three biophysical parameters, especially $\varepsilon_L$ and $\gamma$, are often perturbed

**Table 2.** Bayesian inference results with different prior distributions of $\gamma$.

| Mutant | $\epsilon_L^1$ | $\epsilon_L^2$ | $\gamma^1$ | $\gamma^2$ |
|---|---|---|---|---|
| WT | $6.61_{6.51}^{6.73}$ | $6.62_{6.51}^{6.73}$ | $5.29_{5.20}^{5.38}$ | $5.29_{5.20}^{5.38}$ |
| Q32A-E147G | $7.29_{7.02}^{7.53}$ | $7.27_{7.02}^{7.53}$ | $2.45_{2.38}^{2.52}$ | $2.45_{2.38}^{2.52}$ |
| R49G | $8.62_{8.40}^{8.84}$ | $8.62_{8.42}^{8.85}$ | $1.66_{1.60}^{1.72}$ | $1.65_{1.59}^{1.71}$ |
| D53H | $5.46_{5.09}^{5.80}$ | $5.46_{5.05}^{5.79}$ | $1.62_{1.57}^{1.66}$ | $1.62_{1.57}^{1.67}$ |
| P105M | $7.23_{6.70}^{7.77}$ | $7.24_{6.59}^{7.76}$ | $0.92_{0.86}^{0.98}$ | $0.92_{0.86}^{0.98}$ |
| Y132A | $2.35_{2.15}^{2.55}$ | $2.36_{2.17}^{2.55}$ | $6.23_{5.97}^{6.54}$ | $6.22_{5.96}^{6.55}$ |
| G143M | $6.92_{6.67}^{7.18}$ | $6.91_{6.68}^{7.15}$ | $1.52_{1.47}^{1.56}$ | $1.52_{1.47}^{1.56}$ |
| E150Y | $6.39_{6.01}^{6.73}$ | $6.38_{5.99}^{6.73}$ | $1.72_{1.66}^{1.79}$ | $1.72_{1.66}^{1.78}$ |
| PIF | $11.05_{10.92}^{11.17}$ | $11.05_{10.90}^{11.18}$ | $1.54_{1.49}^{1.59}$ | $1.54_{1.49}^{1.59}$ |
| C203V | $4.51_{4.27}^{4.74}$ | $4.53_{4.27}^{4.78}$ | $6.95_{5.89}^{10.23}$ | $7.47_{6.01}^{15.84}$ |
| G102D-T26A | $7.23_{7.09}^{7.36}$ | $7.22_{7.11}^{7.35}$ | $4.31_{4.16}^{4.51}$ | $4.31_{4.15}^{4.52}$ |
| G102D-Y42M-I57N | $8.21_{8.09}^{8.34}$ | $8.21_{8.09}^{8.34}$ | $2.83_{2.78}^{2.89}$ | $2.83_{2.78}^{2.88}$ |
| G102D-K98Q | $5.96_{5.51}^{6.36}$ | $5.97_{5.55}^{6.37}$ | $2.14_{2.01}^{2.28}$ | $2.14_{2.00}^{2.29}$ |
| G102D-L146A | $5.70_{5.51}^{5.88}$ | $5.70_{5.52}^{5.88}$ | $5.43_{5.03}^{6.05}$ | $5.42_{5.03}^{6.05}$ |
| G102D-HQQ | $7.56_{7.30}^{7.83}$ | $7.57_{7.23}^{7.86}$ | $1.54_{1.48}^{1.61}$ | $1.54_{1.48}^{1.60}$ |
| Y132A-G102D-T26A | $9.55_{8.77}^{10.25}$ | $9.67_{8.92}^{10.51}$ | $-2.96_{-3.39}^{-2.58}$ | $-3.02_{-3.47}^{-2.66}$ |
| Y132A-R49G | $5.77_{5.56}^{5.98}$ | $5.78_{5.58}^{5.98}$ | $3.11_{3.05}^{3.16}$ | $3.11_{3.05}^{3.16}$ |
| Y132A-PIF | $8.71_{8.48}^{8.93}$ | $8.70_{8.47}^{8.93}$ | $3.69_{3.57}^{3.84}$ | $3.69_{3.57}^{3.83}$ |
| Y132A-C203V | $4.14_{3.94}^{4.32}$ | $4.15_{3.96}^{4.34}$ | $7.82_{6.46}^{11.12}$ | $9.30_{6.76}^{16.72}$ |
| C203V-R49G | $5.40_{5.14}^{5.63}$ | $5.39_{5.13}^{5.66}$ | $2.38_{2.33}^{2.43}$ | $2.38_{2.33}^{2.44}$ |
| C203V-D53H | $5.26_{4.98}^{5.53}$ | $5.26_{4.97}^{5.51}$ | $1.42_{1.39}^{1.46}$ | $1.43_{1.39}^{1.46}$ |
| C203V-G102D-L146A | $3.13_{2.81}^{3.43}$ | $3.16_{2.78}^{3.49}$ | $6.74_{6.00}^{9.01}$ | $6.99_{6.03}^{13.63}$ |
| C203V-PIF | $8.30_{7.95}^{8.61}$ | $8.30_{7.94}^{8.65}$ | $3.49_{3.36}^{3.64}$ | $3.49_{3.35}^{3.64}$ |

The column of $p^1/p^2$ shows the Bayesian inference results using the Gaussian prior distribution of $\gamma$ centered at 5 $k_BT$ with a standard deviation of 2.5/5 $k_BT (p = \epsilon_L$ or $\gamma)$. The numbers in the table are the medians of the inferred posterior distributions for the corresponding parameters, with their superscripts/subscripts labeling the upper/lower bound of the 95% credible regions (estimated from 1000 posterior samples).

together in the mutants. Indeed, although the three parameters are theoretically independent of each other, at the structural level, it is less likely to have a scenario where a mutation perturbs inter-domain coupling ($\gamma$) but leaves intra-domain properties unchanged. Lastly, while quantification of G102D from its flat induction curve incurs large uncertainties (*Figure 2—figure supplement 3*), the diversities of the induction curves and the biophysical parameters of the G102D-rescuing mutants (*Figure 3B*) provide clear support for the qualitative anticipation from the previous section; i.e., rescuing mutations of a dead mutant may restore inducibility through different combinations of tuning $\varepsilon_D$, $\varepsilon_L$, and

$\gamma$. For example, while Y42M-I57N and K98Q both rescue G102D to achieve similar $FC^{1000}$, they exert distinct influences on $\varepsilon_D$ and $\varepsilon_L$, as reflected in the variations of leakiness and $EC_{50}$ between the corresponding induction curves. In another case, while the induction curves of G102D-T26A and G102D-L146A show comparable values of saturation and leakiness, their different $EC_{50}$ values suggest the different effects of the two rescuing mutations on $\varepsilon_L$, further quantified by the inferred parameter values. The diverse mechanistic origins of the rescuing mutations revealed here provide a rational basis for the broad distributions of such mutations. Integrating such thermodynamic analysis with structural and dynamic assessment of allosteric proteins for efficient and quantitative rescuing mutation design could present an interesting avenue for future research, particularly in the context of biomedical applications (*Pan et al., 2005*; *Liu and Nussinov, 2008*).

In summary, *Equation 1* accurately captures the induction curves of a comprehensive set of TetR mutants using a minimum set of parameters with realistic values (see the section Discussion and *Figure 3—figure supplement 1*). The two-domain model thus provides a quantitative platform for investigating TetR allostery beyond an intuitive rationalization of the DMS results.

## Exploring epistasis between mutations

With the 15 mutants mapped to the parameter space of the two-domain model, we now explore the epistatic interactions between the relevant mutations. To do so, we start by assuming additivity in the perturbation of all three biophysical parameters (see *Equations 4 and 5*, where $p$ represents any one of $\varepsilon_D$, $\varepsilon_L$, and $\gamma$).

$$p^{mut1+mut2} - p^{WT} = \alpha_{1,p}\delta p^{mut1} + \alpha_{2,p}\delta p^{mut2} \tag{4}$$

$$\delta p^{mut1} = p^{mut1} - p^{WT} \tag{5}$$

We then evaluate how the induction curves generated by this additive model ($\alpha_{1,p} = \alpha_{2,p} = 1$) deviate from the corresponding experimental results, and the magnitude of deviation quantifies the significance of epistasis. Eight mutation combinations were chosen for the analysis, where we pair up C203V and Y132A, the two single mutations that enhance the level of induction relative to the WT, with mutations from different structural regions of TetR (see Appendix 1 section 'Mutation selection for two-domain model analysis' for more discussions).

Although predictions from the additive model are qualitatively correct for five of the eight mutation combinations on phenotypic effects, none of them captures the corresponding induction curve accurately (*Figure 4—figure supplement 1*). This again highlights the parameter space degeneracy of the phenotypes, and that mechanistic specificity is required of a model for making reliable predictions on combined mutants (*Li and Lehner, 2020*). For a deeper understanding of epistatic interactions between the mutations queried here, we next directly fit for the three biophysical parameters of the combined mutants using their induction curves, which are then compared with those obtained from the additive model.

As shown in *Figure 4—figure supplement 2A*, the induction data of all eight combined mutants are well captured by the fitted curves, which reaffirms the applicability of the two-domain model. Interestingly, despite the discrepancy between the experimental induction curves and those predicted by the additive model, the fitted parameters of the combined mutants are comparable to those from the additive model in many cases (see *Figure 4—figure supplement 2B*). In this regard, one noticeable example is the mutant C203V-G102D-L146A, for which the result of direct fitting is close to the additive model in terms of both $\varepsilon_L$ and $\gamma$ (with a difference of $\leqslant 0.5\ k_BT$), while they differ more in $\varepsilon_D$ (1.6 $k_BT$). Inspired by such observations, we then seek the minimal modifications to the additive model that can account for epistasis. As C203 is one of the most distant residues from DBD in TetR, we reason that its effect on $\varepsilon_D$ may get dominated by mutations much closer to the DNA-binding residues like G102D and L146A (*Figure 3—figure supplement 1* and *Table 1*). Remarkably, when we quench C203V's contribution to $\varepsilon_D$ in the additive model of C203V-G102D-L146A ($\alpha_{C203V,\varepsilon_D} = 0$, see *Equation 4*), the predicted induction curve well recapitulates the experimental data (*Figure 4G*). In another example, when C203V is combined with the triple mutations PIF, which are much closer to the DBD, the fitted parameters of the combined mutant also align well with the additive model except only for $\varepsilon_D$. Here, quenching C203V's contribution to $\varepsilon_D$ again leads to good agreement between the additive model and experiment in terms of the induction curve (*Figure 4H*).

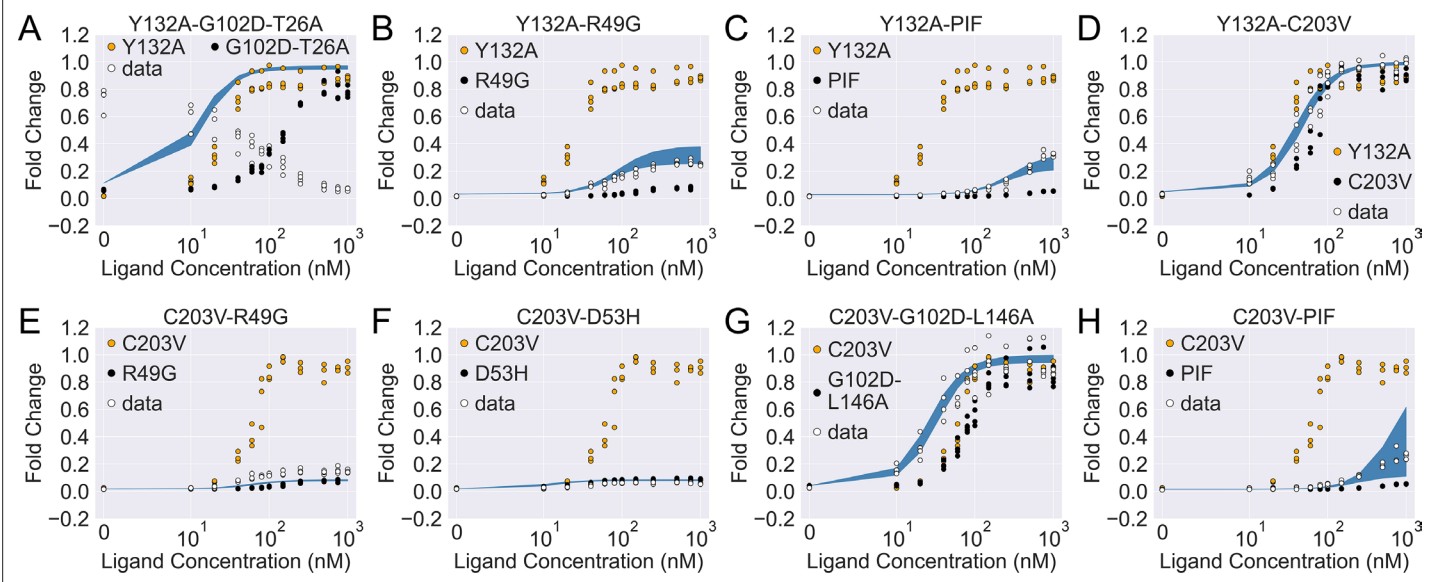

**Figure 4.** The induction curves for the eight combined TetR mutants from experimental measurement and prediction from the modified additive model. In each plot, black, orange, and white points represent the experimental data for mutant 1, mutant 2, and the combined mutant (named mutant 1-mutant 2), specified in the legend and title. The blue band shows the 95th percentile of the induction curve prediction from the modified additive model. The modification to the basic additive model in each plot is specified by the six weights $\{\alpha_{1,\varepsilon_D}, \alpha_{2,\varepsilon_D}, \alpha_{1,\varepsilon_L}, \alpha_{2,\varepsilon_L}, \alpha_{1,\gamma}, \alpha_{2,\gamma}\}$ (see *Equation 4*), which are (**A**) $\{1, 1, 1, 1, 1, 1\}$; (**B**) $\{1, 1, 0.5, 1, 1, 1\}$; (**C**) $\{1, 1, 0.5, 1, 1, 1\}$; (**D**) $\{1, 1, 0, 1, 1, 1\}$; (**E**) $\{0, 1, 1, 1, 0, 1\}$; (**F**) $\{0, 1, 1, 1, 0, 1\}$; (**G**) $\{0, 1, 1, 1, 1, 1\}$; (**H**) $\{0, 1, 1, 1, 1, 1\}$.

The online version of this article includes the following figure supplement(s) for figure 4:

**Figure supplement 1.** The induction curves of the eight combined mutants calculated using the basic additive model (i.e. with $\alpha_{1,p} = \alpha_{2,p} = 1$ in main text *Equation 4*).

**Figure supplement 2.** Induction curves of the eight combined mutants and the corresponding parameter estimation results as well as the basic additive model predictions.

The epistatic effects in other C203V-containing combined mutants can be largely accounted for by modifying the additive model following the same physical reasoning as above. For instance, mutation D53H is near the DNA-binding residues and located at the domain interface, which suggests its dominant role in defining the DBD energetics and inter-domain cooperativity when paired up with C203V (*Scholz et al., 2004*). Accordingly, quenching C203V's effect on both $\varepsilon_D$ and $\gamma$ in the additive model for mutant C203V-D53H leads to dramatic improvement in the induction curve prediction (*Figure 4—figure supplement 1F* and *Figure 4F*). Along this line, success is also observed when accounting for the epistasis between C203V and R49G by the same approach (*Figure 4E*). These examples might also explain why C203V, although being able to enhance the induction of TetR, fails to rescue a range of dead mutations including R49A, D53V, G102D, N129D, and G196D (*Leander et al., 2020*). More broadly, these observations together indicate that epistasis in combined mutants (e.g. those containing C203V) can be captured by the additive two-domain model with modifications based on physical reasoning.

The epistasis in combined mutants containing Y132A can be understood in a similar manner. Noticeably, the fitted parameter values of Y132A-C203V agree very well with the additive model for $\varepsilon_D$ and $\gamma$ (within a difference of 0.4 $k_BT$), while they differ in $\varepsilon_L$ by 3.8 $k_BT$. Interestingly, such discrepancy in $\varepsilon_L$ values can be essentially resolved by quenching Y132A's contribution in the additive model (reduced to 0.4 $k_BT$). This likely suggests that the effect of Y132A on $\varepsilon_L$ tends to be compromised when combined with other mutations that perturb $\varepsilon_L$. Indeed, for most of the Y132A containing mutants investigated here (Y132A-C203V/PIF/R49G), the accuracy of induction curve prediction by the additive model improves greatly when $\alpha_{Y132A,\varepsilon_L}$ is tuned down (*Figure 4B–D*, see Appendix 1 for more discussions).

The only exception to this trend is the mutant Y132A-G102D-T26A, for which we observe strong epistasis between Y132A and the dead-rescue mutation pair G102D-T26A. Although both mutations

Y132A and G102D-T26A enhance the allosteric response of TetR (*Figure 3*), their combined effects radically change the sign of cooperativity between the two domains ($\gamma$), turning the ligand (aTC) from an inducer into a corepressor (*Figure 4A* and *Figure 4—figure supplement 2B*). Here, large disparity exists between the additive model and the direct fitting result in all three biophysical parameters, which cannot be explained by simple modifications of the additive model.

## Discussion

Allostery, a major regulatory mechanism in biology, has attracted intense research interest in the past few decades due to its complexity and implications in biomedicine and protein engineering. A central goal is to develop a quantitative understanding of the phenomena through a physical model that can be tested by comprehensive data. Along this line, one of us has in recent years advanced a function-centric approach to studying protein allostery with DMS, which provides a comprehensive and most direct test of our mechanistic understandings. Specifically, we have shown in an unbiased way that allostery hotspots and dead-rescue mutation pairs in four TetR family TFs are distributed across the protein structures (*Leander et al., 2022*; *Leander et al., 2020*). This highlights that modifying the propagation of conformational distortions between the effector and active sites is not the only way to tune allosteric regulation (*Leander et al., 2020*).

The rich and surprising observations for the TetR family TFs call for a physical understanding of the underlying allostery mechanism, for which we resort to statistical thermodynamic models. Statistical thermodynamic models have played a central role in shaping our understanding of allostery (*Monod et al., 1965*; *Koshland et al., 1966*). In particular, *Chure et al., 2019* have developed the methodology of fitting the MWC model to the induction data of LacI repressors using Bayesian inference. This established the connection between sequence-level perturbations to system-level responses, which enables the exploration of mutational effects on transcription within the framework of biophysical models. However, as pointed out in several previous studies, the MWC model is not consistent with the observation that effector-bound TetR crystal structures are closer to the DNA-bound form compared to the apo crystal structures (*Motlagh et al., 2014*; *Reichheld et al., 2009*; *Hilser et al., 2012*). Hence, it's at least difficult to provide a complete picture of TetR allostery with the MWC model. The EAM on the other hand presents a more flexible and detailed domain-specific view of allostery, which can be applied to more complex allosteric systems (e.g. allostery with intrinsically disordered proteins) (*Motlagh et al., 2014*; *Hilser et al., 2012*).

Inspired by these pioneering developments and our DMS results, in this work, we propose a two-domain statistical thermodynamic model, the parameter-activity degeneracy of which readily elucidates the distributed allosteric network observed in the DMS result of TetR (*Figure 1*). On the other hand, the functional form derived in *Equation 1* establishes the qualitative differences in the impacts of various model parameters on the characteristics of the induction curve (*Figure 2*); e.g., dead mutations perturbing inter- and intra-domain properties are predicted to cause induction curves with and without a saturating plateau, which are both observed experimentally (*Figure 3*). Moreover, *Equation 1* accurately captures the induction data of a diverse set of TetR mutants (*Figure 3*, *Figure 3—figure supplement 1*, and *Figure 4—figure supplement 2*) in a quantitative manner, enabling their mapping to the model parameter space with high precision. It's noted that the homodimeric nature of TetR is ignored in the current two-domain model to minimize the number of parameters, and additional experimental data could necessitate a more complex model for TetR allostery in the future (see Appendix 1 section 'The simplicity of the two-domain model' for more discussions).

The success of *Equation 1* allows for a quantitative investigation of epistasis between the characterized mutations. In a previous study of another TF, LacI, using the MWC model, no epistasis was observed between mutations of ligand-binding and DNA-binding residues analyzed therein (*Chure et al., 2019*). In the present analysis of TetR, epistasis is observed in all eight queried mutation combinations, which can be largely rationalized through physical reasoning. For example, our results intuitively indicate that the effect of a distant mutation like C203V on DBD energetics and inter-domain coupling tend to be dominated by mutations much closer to these regions (*Figure 4* and *Figure 4—figure supplement 1*). Such phenomena suggest that these mutations affect allosteric regulation by influencing not only the relative populations of conformations in different ligation states, but also through shifting the dominant conformations themselves (*Zhang et al., 2020*). On the other hand, epistasis involving Y132A is more complex. For instance, we observe that Y132A's influence on $\varepsilon_L$ is

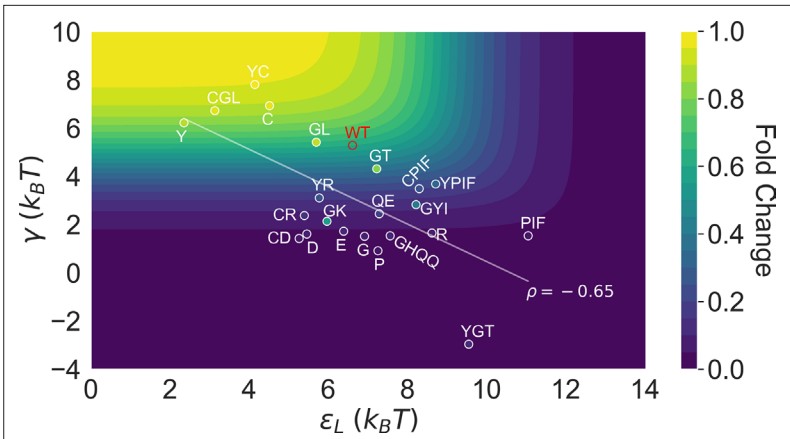

**Figure 5.** Distribution of the 23 investigated TetR mutants in the parameter space of the two-domain model illustrates the correlation between perturbations in $\varepsilon_L$ and $\gamma$. Color of the contour plot encodes the fold change at $c = 1000$ nM calculated for each point in the two-dimensional space of $\varepsilon_L$ and $\gamma$ with the wildtype (WT) $\varepsilon_D$ value. The color within the data point of each mutant is based on the $FC^{1000}$ calculated with its specific $\varepsilon_D$ value. The notation of each mutant is abbreviated based on the one-letter codes of the residues that are mutated. The specific mutations corresponding to each letter code from upper left to lower right are C: C203V; Y: Y132A; GL: G102D-L146A; GT: G102D-T26A; R: R49G; D: D53H; GK: G102D-K98Q; E: E150Y; QE: Q32A-E147G; G: G143M; P: P105M; GYI: G102D-Y42M-I57N; GHQQ: G102D-H44F-Q47S-Q76K; PIF: P176N-I174K-F177S. The least-squares regression line between $\varepsilon_L$ and $\gamma$ is shown with their Pearson correlation coefficient ($\rho$).

diminished when paired up with other mutations in most cases. Further mechanistic understanding, likely from a biochemical perspective, is required to fully explain this phenomenon. Nonetheless, these observations provide a plausible explanation for why C203V and Y132A, although being able to enhance the induction response of TetR individually, could not rescue a range of dead mutations (**Leander et al., 2020**).

Our results reveal additional insights into the nature of two-domain allostery as well. Although our model assumes no a priori correlation between the intra- and inter-domain properties of the TF, we find that they are always modified together by mutations (**Figure 3** and **Figure 4—figure supplement 2**). This immediately points to the interconnectivity of allosteric networks (**Takeuchi et al., 2019**; **Reichheld et al., 2009**; **Scholz et al., 2004**). That is, when a residue is involved in the conformational rearrangements induced by both effector and operator binding, it functions as a bridge through which the two events are coupled. Within such a conceptual framework, it is highly unlikely to observe a mutation that changes $\gamma$ alone without perturbing the intra-domain energetics.

This point is explicitly illustrated in **Figure 5**, which summarizes the locations of all the characterized 23 mutants in the two-dimensional space of $\varepsilon_L$ and $\gamma$. Here, the color of the contour plot encodes the $FC^{1000}$ value calculated for each ($\varepsilon_L$, $\gamma$) combination together with the WT $\varepsilon_D$ value, while the color of the specific mutant data points is based on the true $FC^{1000}$ value. Thus, the color of a data point reflects the $\varepsilon_D$ of the corresponding mutant relative to the WT; i.e., a brighter/darker color than the surrounding indicates that the mutant features a higher/lower $\varepsilon_D$ value than the WT. Since the variations of $\varepsilon_D$ among the investigated mutants are modest compared with $\varepsilon_L$ and $\gamma$ (see **Figure 3B**), we'll primarily focus on the latter two in the Discussion. Evidently, mutants with the strongest allosteric responses feature small $\varepsilon_L$ and large $\gamma$, which correspond to the upper left region of **Figure 5**. Additional mutations in the background of these mutants lead to weaker allosteric response, moving the mutants to the darker regions of the plot. The interconnected nature of allosteric networks described by the two-domain model, however, dictates that such shifts would take place along the diagonal of the $\varepsilon_L$-$\gamma$ plane, which describe mutations that modify intra- and inter-domain properties simultaneously. This feature potentially offers a functional advantage of preventing mutants from regions of low allosteric response (corresponding to upper right and lower left of **Figure 5**), facilitating efficient adaptation to new effectors during evolution. In the future, it is of interest to examine whether such a negative correlation between $\varepsilon_L$ and $\gamma$ is a generic feature of two-domain allostery.

Our results establish the two-domain model as a flexible and quantitative platform for investigating TetR allostery, adding to the value of statistical thermodynamic models as advocated by the seminal works on MWC molecules (*Daber et al., 2011*; *Chure et al., 2019*). We note, however, that the two-domain model is a more generic description of allostery compared with the MWC model.

First, although our model identifies the importance of intra-domain energetics in tuning the free energies of different TF states, it points out the more fundamental role of inter-domain coupling in defining allosteric response (*Figure 2* and Appendix 1). A large inter-domain coupling is assumed to be obligatory in the MWC model and required to create the two pre-existing conformations of different ligand and operator affinities in the first place. Various experimental studies as well as our own data reveal that the inter-domain coupling of TetR is amenable to tuning, especially by mutations of residues at the domain interface (*Reichheld et al., 2009*; *Scholz et al., 2004*; *Müller et al., 1995*; *Hecht et al., 1993*). In some cases, even single mutations can reverse the sign of cooperativity between LBD and DBD, indicating that the dead mutational effects observed in the DMS experiments and elsewhere could well originate from diminishing $\gamma$. These considerations necessitate an explicit treatment of $\gamma$ in a physical model of TetR allostery.

Second, the MWC model, when applied to two-domain systems, contains at least four biophysical parameters (*Daber et al., 2011*; *Chure et al., 2019*), is thus more complex than our two-domain model, which requires only three. The simplicity of our model reduces its parameter space degeneracy, enabling high precision in fitting while being able to accurately describe a broad set of induction data. The three parameters also offer an interpretable mechanistic picture of TetR allostery in a physically transparent fashion.

We have focused our experimental and theoretical analyses on the TetR family TFs, which represent important systems for building allosteric models, due to their broad involvement in many aspects of cell physiology (*Ramos et al., 2005*; *Cuthbertson and Nodwell, 2013*). Along this line, a natural extension of the current study is to use the two-domain model as a platform for comparing allostery in different TetR homologs (*Leander et al., 2022*). For example, coupling the model with systematic dose-response study of hotspot mutations could reveal the roles of different allosteric hotspots (whether they dictate intra- or inter-domain properties or both). The similarity in key structural features among TetR homologs then enables the comparison of the nature of hotspots at similar structural locations. This could potentially lead to novel insights into the different roles that sequence and structure play in defining the allostery of a system, which is a fundamental question to address in rational engineering of allosteric proteins.

Although our two-domain model is inspired by and established with the data of TetR family TFs, the mechanistic and modeling insights gained here should be generally applicable to other allosteric systems sharing the two-domain architecture. This includes other TFs like CAP, a homodimeric TF with a cyclic AMP (effector) binding domain and a DNA-binding domain (*Tzeng and Kalodimos, 2012*); receptors like pLGICs, which allosterically translate the binding of neurotransmitters to their extracellular domains into the activation of ionotropic pores located in their transmembrane domains (*Sauguet et al., 2015*; *Hu et al., 2020*); and allosteric enzymes like ATCase, in which binding of effector like ATP to the regulatory domain allosterically alters the catalytic activity of the functional domain (*Velyvis et al., 2007*; *Lipscomb and Kantrowitz, 2012*). Ultimately, we envision that such model provides a minimalist but physically sound framework for integrating different types of experimental data and computational methods (*Leander et al., 2022*; *Yuan et al., 2022*; *Xie et al., 2023*; *Tonner et al., 2022*;) for exploring rational engineering and regulation of allostery for biotechnological and biomedical applications.

# Materials and methods
## Library construction
Using a low-copy backbone (SC101 origin of replication) carrying spectinomycin resistance, we constructed a sensor plasmid with TetR(B) (Uniprot P04483). The *tetRb* gene was driven by a variant of promoter apFAB61 and Bba J61132 RBS (*Kosuri et al., 2013*). On a second reporter plasmid, superfolder GFP (*Pédelacq et al., 2006*) was cloned into a high-copy backbone (ColE1 origin of replication) carrying kanamycin resistance. The expression of the superfolder GFP reporter was placed under the control of the ptetO promoter. To control for plasmid copy number, RFP was constitutively expressed

with the BBa J23106 promoter and Plotkin RBS (*Kosuri et al., 2013*) in a divergent orientation to sfGFP.

## Library synthesis

A comprehensive single-mutant TetR library was generated by replacing WT residues at positions 2–207 of TetR to all other 19 canonical amino acids (3914 total mutant sequences). Oligonucleotides encoding each single-point mutation were synthesized as single-stranded Oligo Pools from Twist Bioscience and organized into six subpools, spanning six segments of the *tetRb* gene, corresponding to residues 2–39, 40–77, 78–115, 116–153, 154–191, and 192–207 of TetR(B), respectively. Additional sequence diversity was observed in the library due to error rates in the synthesis of single-stranded Oligo Pools, leading to the downstream identification of some double and triple mutant TetR variants. Oligo Pools were encoded as a concatemer of the forward priming sequence, a BasI restriction site (5′-GGTCTC), six-base upstream constant region, TetR mutant sequence, six-base downstream constant region, a BsaI site (5′-GAGACC), and the reverse priming sequence. Subpools were resuspended in double-distilled water (ddH$_2$O) to a final molal concentration of 25 ng/µL and amplified using primers specific to each oligonucleotide subpool with KAPA SYBR FAST qPCR (KAPA Biosystems; 1 ng template). A second PCR amplification was performed with KAPA HiFi (KAPA Biosystems; 1 µL qPCR template, 15 cycles maximum). We amplified corresponding regions of pSC101_TetR_specR with primers that linearized the backbone, added a BsaI restriction site, and removed the replaced WT sequence. Vector backbones were further digested with DpnI, BsaI, and Antarctic phosphatase before library assembly.

We assembled mutant libraries by combining the linearized sensor backbone with each oligo subpool at a molar ratio of 1:5 using Golden Gate Assembly Kit (New England Biolabs; 37°C for 5 min and 60°C for 5 min, repeated 30 times). Reactions were dialyzed with water on silica membranes (0.025 µm pores) for 1 hr before transformed into DH10B cells (New England Biolabs). Library sizes of at least 100,000 colony-forming units (cfu) were considered successful. DH10B cells containing the reporter pColE1_sfGFP_RFP_kanR were transformed with extracted plasmids to obtain cultures of at least 100,000 cfu. Following co-transformation, the cultures for each subpool were stored as glycerol stocks and kept at –80°C.

## Fluorescence-activated cell sorting

The subpool library cultures were seeded from glycerol stocks into 3 mL lysogeny broth (LB) containing 50 µg/mL kanamycin (kan) and 50 µg/mL spectinomycin (spec) and grown for 16 hr at 37°C. Each culture was then back-diluted into two wells of a 96-well plate containing LB kan/spec media using a dilution factor of 1:50 and grown for a period of 5 hr at 37°C. Following incubation, each well containing saturated library culture was diluted 1:75 into 1× phosphate saline buffer (PBS), and fluorescence intensity was measured on an SH800S Cell Sorter (Sony). We first gated cells to remove debris and doublets and selected for variants constitutively expressing RFP. Using this filter, we then proceeded to draw an additional gate to select for mutants that displayed low fluorescence intensities in the absence of aTC. This gate allowed us to select for mutants which retained the ability to repress GFP expression in the presence of an inducer, using prior measurements of WT TetR(B) as a reference for selecting repression competent mutants (*Leander et al., 2020*). Utilizing these gates, we sorted 500,000 events for each gated population and recovered these cells in 5 mL of LB at 37°C before adding 50 µg/mL of kan and spec. Following the addition of antibiotics, the cultures were incubated at 37°C for 16 hr.

Following overnight growth, each culture was back-diluted into two wells of a 96-well plate containing LB kan/spec media using a 1:50 dilution factor. Upon reaching an OD600 ~ 0.2, aTC was added to one of the two wells at a final concentration of 1 µM, with the well not receiving aTC serving as the uninduced control population. Cells were then incubated for another 4 hr growth period at 37°C before being diluted into 1× PBS using a dilution factor of 1:75, and fluorescence intensity was measured on an SH800S Cell Sorter.

We first gated cells to remove debris and doublets and selected for variants constitutively expressing RFP before obtaining the distribution of fluorescence intensities (FITC-A) across the uninduced and induced populations of each of the six subpools (*Figure 3—figure supplement 8*). Using the uninduced control population of each subpool as a reference, gates were drawn to capture mutants which

displayed no response to the presence of aTC in the induced populations. Using these gates, 500,000 events were sorted from the induced populations of each subpool and cells were subsequently recovered at 37°C using 5 mL of LB media before being plated onto LB agar plates containing 50 µg/mL kan and spec.

### Clonal screening of dead mutants

To screen for functionally deficient variants of TetR, 100 individual colonies were picked from each subpool and grown to saturation in a 96-well plate for 6 hr. Saturated cultures were then diluted 1:50 in LB kan/spec media and grown in the presence and absence of 1 µM aTC for 6 hr before OD600 and GFP fluorescence (gain: 40; excitation: 488/20; emission: 525/20) were read on a multiplate reader (Synergy HTX, BioTek). Fluorescence was normalized to OD600, and the fold-inductions of each clone were calculated by dividing its normalized fluorescence in the presence of inducer by the normalized fluorescence in the absence of inducer. The fold-induction of WT TetR was tested in parallel with these screens to serve as a benchmark for function, and clones which displayed <50% activity to WT were replated and validated in triplicate before being sent for Sanger sequencing (Functional Biosciences).

### aTC dose-response measurements

TetR mutants identified during clonal screening were reinoculated into 3 mL of LB kan/spec media and grown overnight. After an overnight growth period, each culture was added to 4 rows of 12 wells in a 96-well plate containing LB kan/spec media using a dilution factor of 1:50. Each mutant was tested across 12 different concentrations of aTC in quadruplicate fashion, with each row having an identical concentration gradient across the 12 wells. The aTC concentration gradient across the 12 wells were 0 nM, 10 nM, 20 nM, 40 nM, 60 nM, 80 nM, 100 nM, 150 nM, 250 nM, 500 nM, 750 nM, and 1 M aTC.

Following reinoculation into 96-well plates, selected mutants were placed on a plate shaker set to 900 RPM and incubated at 37°C for 5 hr. Following this 5 hr growth period, the plates were removed from the plate shaker and the OD600 and GFP fluorescence (gain: 40; excitation: 488/20; emission: 525/20) of each well was read using a multiplate reader (Synergy HTX, BioTek). Fluorescence was normalized to OD600 for each well. The normalized fluorescence at each concentration was measured for four replicates of each mutant (unless otherwise stated). The fluorescence of GFP reporter-only control (no TetR), grown under identical conditions, was measured in the same way. This reporter-only control served as an upper bound to the fluorescence that could be measured in the experiment and was accompanied by a WT TetR-GFP reporter control, used to certify the precision of measurements made at different times.

### Synthesis of combined TetR mutants

To assess the predictive capability of the additive model, eight combined mutants were selected to undergo aTC dose-response characterization. These combined mutants were synthesized using clonal DNA fragments from Twist Biosciences coding for the amino acid sequence of each of the selected combined mutants. The gene fragments were individually cloned into sensor plasmid backbones identical to those used in the previous identification of non-functional TetR(B) mutants following the manufacturer's protocol for Gibson Assembly (New England Biolabs). The newly constructed plasmids carrying each of the combined mutants were then transformed into *Escherichia coli* (DH10B) cells containing a GFP reporter plasmid identical to that used in the previous characterization of non-functional TetR(B) mutants. After a 1 hr recovery period, 100 µL of recovered cells were plated onto LB kan/spec agar plates for each of the combination mutants and incubated for 16 hr at 37°C.

Following overnight growth, individual colonies were picked and inoculated into 3 mL of LB kan/spec media to be grown overnight at 37°C. Following overnight growth, samples from each culture were submitted for Sanger sequencing to confirm the identity of each of the synthesized combined mutants. Following sequence verification, the aTC dose-response behavior of each combined mutant was characterized using an experimental setup consistent with dose-response measurements made for the previously identified TetR(B) mutants.

### Model parameter estimation

As described above, the estimation of the three biophysical parameters ($\varepsilon_D$, $\varepsilon_L$, and $\gamma$) for each mutant is divided into two steps. First, as $\varepsilon_L$ and $\gamma$ do not affect $FC$ at $c = 0$ (***Equation 1***), we directly calculate

the value of $\varepsilon_D$ from the leakiness of an induction curve, which is measured with high precision in our experiments. With the calculated $\varepsilon_D$ ($\varepsilon_D{}^{WT}$ is set to 0 as reference), we then fit the full induction curve to obtain the values of $\varepsilon_L$ and $\gamma$ using the method of Bayesian inference. Here, we first construct the statistical model that describes the data-generating process based on *Equation 1*, and specify the prior distributions of the relevant parameters. This enables the derivation of the conditional probability of the parameter values of a mutant given its induction data, known as the posterior distribution. The posterior distribution is then sampled using Markov chain Monte Carlo (MCMC), from which we infer the values of $\varepsilon_L$ and $\gamma$ directly. The validity of the statistical model and computational algorithm is fully tested with several metrics, and all details regarding the parameter estimation process briefed here are provided in Appendix 1 and *Figure 3—figure supplement 2* to *Figure 3—figure supplement 6*.

## Acknowledgements

This work is funded by NIH Director's New Innovator Award DP2GM132682 (SR) and Shaw Scientist Award (SR), and R35-GM141930 (QC). Development of the Bayesian inference model was partially supported by grant ML-21-016 from the Dreyfus foundation (QC). Computational resources from the Extreme Science and Engineering Discovery Environment (XSEDE), which is supported by NSF grant number ACI-1548562, are greatly appreciated; part of the computational work was performed on the Shared Computing Cluster which is administered by Boston University's Research Computing Services (https://www.bu.edu/tech/support/research).

## Additional information

### Competing interests

Qiang Cui: Senior editor, *eLife*. The other authors declare that no competing interests exist.

### Funding

| Funder | Grant reference number | Author |
| --- | --- | --- |
| National Institutes of Health | R35-GM141930 | Qiang Cui |
| Camille and Henry Dreyfus Foundation | ML-21-016 | Qiang Cui |
| National Institutes of Health | DP2GM132682 | Srivatsan Raman |

The funders had no role in study design, data collection and interpretation, or the decision to submit the work for publication.

### Author contributions

Zhuang Liu, Conceptualization, Formal analysis, Investigation, Methodology, Writing - original draft, Writing - review and editing; Thomas G Gillis, Investigation, Writing - original draft; Srivatsan Raman, Conceptualization, Supervision, Funding acquisition, Writing - original draft, Writing - review and editing; Qiang Cui, Conceptualization, Formal analysis, Supervision, Writing - original draft, Writing - review and editing

### Author ORCIDs

Zhuang Liu ⓘ http://orcid.org/0000-0003-4695-7142
Srivatsan Raman ⓘ http://orcid.org/0000-0003-2461-1589
Qiang Cui ⓘ http://orcid.org/0000-0001-6214-5211

Reviewer #1 (Public Review): https://doi.org/10.7554/eLife.92262.3.sa1
Reviewer #2 (Public Review): https://doi.org/10.7554/eLife.92262.3.sa2
Author response https://doi.org/10.7554/eLife.92262.3.sa3

## Additional files

### Supplementary files
• MDAR checklist

### Data availability
All codes used in this work can be accessed at the Github repository: https://github.com/liuzhbu/Two_Domain_Allostery, copy archived at *liuzhbu, 2023*.

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

# Appendix 1

## A parameterized two-domain thermodynamic model explains diverse mutational effects on protein allostery

Derivation of main text Equation 1

For nonzero ligand concentration

*Equation 1* of main text expresses the gene expression level rescaled by that of an unregulated promoter (fold change) as a function of system and allosteric parameters, whose detailed derivation is given below. Following Griffin et al. (*Chure et al., 2019*) and previous works on thermodynamic models of transcription (*Ackers et al., 1982*; *Buchler et al., 2003*; *Vilar and Leibler, 2003*; *Bintu et al., 2005*; *Kuhlman et al., 2007*; *Daber et al., 2011*), the gene expression level is considered to be proportional to the probability of the promoter being bound by RNAP. Accordingly, the possible occupancy states of the promoter and their corresponding statistical weights are listed in main text *Figure 2—figure supplement 1*, from which we can calculate the probability of the RNAP-bound state as its statistical weight divided by the partition function as given in *Equation 6*.

$$p_{RNAP}^{bound} = \frac{\frac{P}{N_{NS}}e^{-\Delta\varepsilon_P}}{1 + \frac{P}{N_{NS}}e^{-\Delta\varepsilon_P} + \frac{R_1}{N_{NS}}e^{-\Delta\varepsilon_{RI}} + \frac{R_2}{N_{NS}}e^{-\Delta\varepsilon_{RI}} + \frac{R_3}{N_{NS}}e^{-\Delta\varepsilon_{RA}} + \frac{R_4}{N_{NS}}e^{-\Delta\varepsilon_{RA}}} \tag{6}$$

Here, $P$ is the average number of RNAP per cell. $R_1$, $R_2$, $R_3$, and $R_4$ are the average numbers of repressors in the $L_I D_I$, $L_A D_I$, $L_I D_A$, and $L_A D_A$ state per cell, respectively. $N_{NS}$ is the number of non-specific DNA-binding sites in the cell, which is taken to be the length of the *E. coli* genome in base pairs ( 4.6x10⁶) (*Chure et al., 2019*). All RNAP and repressors are assumed to be bound to either the specific or non-specific binding sites on DNA (*Chure et al., 2019*; *Garcia and Phillips, 2011*; *Rünzi and Matzura, 1976*; *von Hippel et al., 1974*; *Kao-Huang et al., 1977*; *Normanno et al., 2015*; *Stracy et al., 2021*; *von Hippel and Berg, 1989*), where the specific binding sites of repressor and RNAP are the operator and promoter sequence, respectively (such assumption is not essential for the derivation of main text; *Equation 1*; *Vilar and Leibler, 2003*, see the last part of this section). When a repressor binds to the operator (orange rectangle in main text *Figure 2—figure supplement 1*), it obstructs RNAP from binding to the adjacent promoter (blue rectangle in main text *Figure 2—figure supplement 1*) and the expression of downstream genes (right pink rectangle in main text *Figure 2—figure supplement 1*). $\Delta\varepsilon_P$, $\Delta\varepsilon_{RA}$, and $\Delta\varepsilon_{RI}$ represent the energy differences between specific and non-specific DNA binding of RNAP, repressor with DBD in the active and inactive conformations, respectively. Note that the free energy quantities are all measured in the unit of $k_B T$, thus the Boltzmann factors throughout our discussion do not include $k_B T$ explicitly.

The statistical weight of a given promoter occupancy state is evaluated based on its energy and the number of microscopic states it corresponds to. With the empty promoter taken as the reference state (statistical weight equals 1), we can then calculate the statistical weights of the other operator states as their probabilities relative to the reference state. Since $N_{NS} \gg R$ ($R = \sum_{i=1}^{4} R_i$ ~ 5000 is the average number of all repressors in the cell) and $P$ ($P \approx 1000$) (*Klumpp and Hwa, 2008*; *Kosuri et al., 2013*), the non-specific binding of repressors and RNAP are considered to be independent. When the promoter is occupied by an RNAP, there are $\binom{N_{NS}}{P-1}$ ways to arrange the remaining $(P-1)$ RNAPs on the $N_{NS}$ non-specific binding sites. Thus, the probability of RNAP-bound state relative to the empty promoter state (all $P$ RNAPs are bound to non-specific binding sites) is given by

$$\frac{p_{RNAP}^{bound}}{p^{empty}} = \frac{\binom{N_{NS}}{P-1}}{\binom{N_{NS}}{P}}e^{-\Delta\varepsilon_P}, \tag{7}$$

where the energy difference between the two promoter occupancy states is accounted for in the Boltzmann weight. As $N_{NS} \gg P$, the right-hand side (RHS) of *Equation 7* can be easily simplified to $\frac{P}{N_{NS}}e^{-\Delta\varepsilon_P}$, which is the statistical weight of the RNAP-bound promoter state given in *Equation 6* and main text *Figure 2—figure supplement 1*. The statistical weights of the other repressor-bound states of promoter are calculated following the same reasoning. It's noted that, since the number of promoter-operator sequences present in a cell (~10–20) is much smaller than $P$ and $R$ (see the main text section Materials and methods), the binding status of each promoter is regarded to be independent of others (*Brewster et al., 2014*).

Next, the $p_{RNAP}^{bound}$ of a repressor-regulated promoter is divided by that of an unregulated promoter (where $R=0$) to get the relative gene expression level, which is defined as fold change ($FC$) in main text *Equation 1*.

$$FC = \frac{p_{RNAP}^{bound}(R > 0)}{p_{RNAP}^{bound}(R = 0)} = \frac{1 + \frac{P}{N_{NS}}e^{-\Delta\varepsilon_P}}{1 + \frac{P}{N_{NS}}e^{-\Delta\varepsilon_P} + \frac{R_1 + R_2}{N_{NS}}e^{-\Delta\varepsilon_{RI}} + \frac{R_3 + R_4}{N_{NS}}e^{-\Delta\varepsilon_{RA}}} \tag{8}$$

With the assumption of weak promoter $1 \gg \frac{P}{N_{NS}}e^{-\Delta\varepsilon_P}$, and that the energy difference between specific and non-specific DNA binding of repressor with DBD in the inactive conformation ($\Delta\varepsilon_{RI}$) is small (*Chure et al., 2019*), *Equation 8* can be simplified to

$$FC = \frac{1}{1 + \frac{R_3 + R_4}{N_{NS}}e^{-\Delta\varepsilon_{RA}}} = \frac{1}{1 + \frac{R}{N_{NS}}\left\{p_{R_3}(c) + p_{R_4}(c)\right\}e^{-\Delta\varepsilon_{RA}}}. \tag{9}$$

Here, $p_{R_3}(c)$ and $p_{R_4}(c)$ are the probabilities of a repressor being in the $L_I D_A$ and $L_A D_A$ state at a given ligand concentration $c$, respectively, which can be calculated from the statistical weights of the possible repressor states. As shown in main text *Figure 2—figure supplement 1*, the unbound $L_I D_I$ state is taken as the reference state with a statistical weight of 1. $\varepsilon_L$ ($\varepsilon_D$) is the free energy increase of the repressor when LBD (DBD) transition from the inactive to the active conformation, and $\gamma$ is the additional free energy penalty when both domains adopt the active conformations. $K_A$ and $K_I$ are the dissociation constants of ligand to the repressor with LBD in the active and inactive conformations, respectively.

Similar to the calculation of $p_{RNAP}^{bound}$, the sum of $p_{R_3}(c)$ and $p_{R_4}(c)$ is then given by

$$p_{R_3}(c) + p_{R_4}(c) = \frac{e^{-\varepsilon_D}\left\{1 + \left(\frac{c}{K_I}\right)^2\right\} + e^{-\varepsilon_D - \varepsilon_L - \gamma}\left\{1 + \left(\frac{c}{K_A}\right)^2\right\}}{1 + \left(\frac{c}{K_I}\right)^2 + e^{-\varepsilon_L}\left\{1 + \left(\frac{c}{K_A}\right)^2\right\} + e^{-\varepsilon_D}\left\{1 + \left(\frac{c}{K_I}\right)^2\right\} + e^{-\varepsilon_D - \varepsilon_L - \gamma}\left\{1 + \left(\frac{c}{K_A}\right)^2\right\}}. \tag{10}$$

For the simplicity and mechanistic clarity of the model, we assume that ligand binding to the repressor with LBD in the inactive conformation is negligible ($c/K_I \ll 1$ for $c$ below 1 μM, the maximum ligand concentration used in our experiments). Then plugging *Equation 10* back into *Equation 9*, we have

$$FC = \frac{1}{1 + \frac{Re^{-\Delta\varepsilon_{RA}}}{N_{NS}}e^{-\varepsilon_D}\frac{1 + e^{-\varepsilon_L - \gamma}\left\{1 + \left(\frac{c}{K_A}\right)^2\right\}}{1 + e^{-\varepsilon_L}\left\{1 + \left(\frac{c}{K_A}\right)^2\right\} + e^{-\varepsilon_D} + e^{-\varepsilon_D - \varepsilon_L - \gamma}\left\{1 + \left(\frac{c}{K_A}\right)^2\right\}}}. \tag{11}$$

We can state that $c/K_A \gg 1$ for all ligand concentrations where we measured gene expression levels experimentally (10 nM $\leq c \leq$ 1000 nM) besides $c=0$, based on the apparent dissociation constants of the ligand (aTC) binding to TetR reported in the literature (*Scholz et al., 2000*; *Schubert et al., 2004*) and the assumption that $e^{-\varepsilon_L} \ll 1$. Thus, at the nonzero ligand concentrations where we measured gene expression levels experimentally, *Equation 11* can be simplified to

$$FC = \frac{1}{1 + \frac{Re^{-\Delta\varepsilon_{RA}}}{N_{NS}}e^{-\varepsilon_D}\frac{1 + e^{-\varepsilon_L - \gamma}\left(\frac{c}{K_A}\right)^2}{1 + e^{-\varepsilon_L}\left(\frac{c}{K_A}\right)^2 + e^{-\varepsilon_D} + e^{-\varepsilon_D - \varepsilon_L - \gamma}\left(\frac{c}{K_A}\right)^2}}. \tag{12}$$

Further assuming $e^{-\varepsilon_D} \ll 1$ and $e^{-\varepsilon_D - \gamma} \ll 1$, *Equation 12* is reduced to

$$FC = \cfrac{1}{1 + \cfrac{Re^{-\Delta\varepsilon_{RA}}}{N_{NS}} e^{-\varepsilon_D} \cfrac{1 + e^{-\varepsilon_L - \gamma} \left(\cfrac{c}{K_A}\right)^2}{1 + e^{-\varepsilon_L} \left(\cfrac{c}{K_A}\right)^2}},$$

(13)

which is the same as the main text *Equation 1* after substituting $\frac{Re^{-\Delta\varepsilon_{RA}}}{N_{NS}}$ and $K_A$ with $R^*$ and $K$, respectively, to simplify notations.

All energy terms in the exponents are evaluated in the unit of $k_B T$ in this work, and we've assumed that $e^{-\varepsilon_D}$, $e^{-\varepsilon_D - \gamma}$, $e^{-\varepsilon_L}$, and $e^{-\varepsilon_L - \gamma}$ to be much smaller than one based on the consideration that free energy differences between different conformations of a TetR-like protein is usually on the order of a few $k_B T$ (**Chure et al., 2019**; **Motlagh et al., 2014**).

## For zero ligand concentration

At $c$=0, *Equation 11* becomes

$$FC = \cfrac{1}{1 + \cfrac{Re^{-\Delta\varepsilon_{RA}}}{N_{NS}} e^{-\varepsilon_D} \cfrac{1 + e^{-\varepsilon_L - \gamma}}{1 + e^{-\varepsilon_L} + e^{-\varepsilon_D} + e^{-\varepsilon_D - \varepsilon_L - \gamma}}}.$$

(14)

Under the same set of assumptions stated above, *Equation 14* is reduced to $FC = \left(1 + R^* e^{-\varepsilon_D}\right)^{-1}$, which agrees with *Equation 13* at $c$=0. It should also be noted that $\varepsilon_D$ plays a dominant role in deciding $FC$ at $c$=0 (leakiness of the induction curve) compared with $\varepsilon_L$ and $\gamma$, even based on the full functional form of *Equation 14*. Thus, the difference between the leakiness of the induction curves of different mutants analyzed in this study is attributed solely to their difference in $\varepsilon_D$, as mutations of operator-binding residues are avoided and $R^*$ is taken to be constant across these mutants (see the section 'Model parameter estimation' below).

## Inclusion of single-ligand-bound state of repressor

As shown in the main text section 'Overview of the two-domain thermodynamic model of allostery' and main text *Figure 2—figure supplement 1*, single-ligand-bound state of the repressor is ignored in our symmetric two-domain model and the derivation of *Equation 13* (main text (*Equation 1*) ) for simplicity. Nonetheless, including single-ligand-bound repressor states in the derivation above leads to the same form of *Equation 13*, which we demonstrate below. Allowing single ligand binding, *Equation 10* becomes

$$p_{R_3}(c) + p_{R_4}(c) = \cfrac{e^{-\varepsilon_D}\left(1 + \frac{c}{K_I}\right)^2 + e^{-\varepsilon_D - \varepsilon_L - \gamma}\left(1 + \frac{c}{K_A}\right)^2}{\left(1 + \frac{c}{K_I}\right)^2 + e^{-\varepsilon_L}\left(1 + \frac{c}{K_A}\right)^2 + e^{-\varepsilon_D}\left(1 + \frac{c}{K_I}\right)^2 + e^{-\varepsilon_D - \varepsilon_L - \gamma}\left(1 + \frac{c}{K_A}\right)^2},$$

(15)

while *Equations 6–9* stay unchanged. Accordingly, *Equation 11* becomes

$$FC = \cfrac{1}{1 + \cfrac{Re^{-\Delta\varepsilon_{RA}}}{N_{NS}} e^{-\varepsilon_D} \cfrac{1 + e^{-\varepsilon_L - \gamma}\left(1 + \frac{c}{K_A}\right)^2}{1 + e^{-\varepsilon_L}\left(1 + \frac{c}{K_A}\right)^2 + e^{-\varepsilon_D} + e^{-\varepsilon_D - \varepsilon_L - \gamma}\left(1 + \frac{c}{K_A}\right)^2}},$$

(16)

which is reduced to the same form of *Equation 12* and *Equation 14* at $c/K_A \gg 1$ and $c = 0$, respectively. The subsequent steps in the derivation that lead to *Equation 13* then stay the same as before.

## Alternative derivation of main text Equation 1

*Equation 1* of main text can be derived from a different perspective of the system as well, which is based on the equilibria among different conformational and binding states of the repressor as specified in main text (*Figure 2—figure supplement 2*; *Daber et al., 2011*). As gene expression level is considered proportional to the probability that a promoter is bound by RNAP, it can be alternatively written as

$$FC = \frac{p_{RNAP}^{bound}(R > 0)}{p_{RNAP}^{bound}(R = 0)} = \frac{[O]}{[O] + [R_1O] + [R_2O] + [R_3O] + [R_4O]} \tag{17}$$

Here, $[O]$ is the concentration of free operator in the cell, and $[R_1O]$ to $[R_4O]$ are the concentrations of operator bound repressor that is in the corresponding conformations (main text *Figure 2—figure supplement 2*) regardless of ligand-binding states (e.g. $[R_1O]$ is the total concentration of species $[R_1O]$, $[R_1LO]$, and $[R_1L_2O]$). Based on the equilibrium constants given in main text *Figure 2—figure supplement 2*, we then have

$$FC = \frac{1}{1 + ([R_1] + [R_2])e^{-\varepsilon_{R_IO}} + ([R_3] + [R_4])e^{-\varepsilon_{R_AO}}}, \tag{18}$$

where $[R_1]$ to $[R_4]$ are the concentrations of the repressor in the corresponding conformations divided by 1 M (not bound to operator) regardless of its ligand-binding states. Under the condition that the total number of repressors is much larger than that of operators (or promoters) in the cell ($[R_{tot}] \gg [O_{tot}]$), *Equation 18* can be rewritten in a more familiar form as given below.

$$FC = \frac{1}{1 + [R_{tot}]e^{-\varepsilon_{R_AO}} \{f + s(1 - f)\}} \tag{19}$$

$$f = \frac{e^{-\varepsilon_D}\left(1 + \dfrac{c}{K_I}\right)^2 + e^{-\varepsilon_D - \varepsilon_L - \gamma}\left(1 + \dfrac{c}{K_A}\right)^2}{\left(1 + \dfrac{c}{K_I}\right)^2 + e^{-\varepsilon_L}\left(1 + \dfrac{c}{K_A}\right)^2 + e^{-\varepsilon_D}\left(1 + \dfrac{c}{K_I}\right)^2 + e^{-\varepsilon_D - \varepsilon_L - \gamma}\left(1 + \dfrac{c}{K_A}\right)^2} \tag{20}$$

$$s = \frac{e^{-\varepsilon_{R_IO}}}{e^{-\varepsilon_{R_AO}}} \tag{21}$$

Assuming that the operator binding affinity of a repressor with active DBD is much higher than that of a repressor with inactive DBD ($s \ll 1$), such that the effect of inactive operator binding on gene expression is negligible, *Equation 19* can be further simplified to

$$FC = \frac{1}{1 + [R_{tot}]e^{-\varepsilon_{R_AO}} \dfrac{e^{-\varepsilon_D}\left(1 + \dfrac{c}{K_I}\right)^2 + e^{-\varepsilon_D - \varepsilon_L - \gamma}\left(1 + \dfrac{c}{K_A}\right)^2}{\left(1 + \dfrac{c}{K_I}\right)^2 + e^{-\varepsilon_L}\left(1 + \dfrac{c}{K_A}\right)^2 + e^{-\varepsilon_D}\left(1 + \dfrac{c}{K_I}\right)^2 + e^{-\varepsilon_D - \varepsilon_L - \gamma}\left(1 + \dfrac{c}{K_A}\right)^2}}. \tag{22}$$

With the same assumption used in the derivation from *Equations 10 to 11* in the last section ($c/K_I \ll 1$ for $c$ below 1 µM, the maximum ligand concentration used in our experiments), we have

$$FC = \frac{1}{1 + [R_{tot}]e^{-\varepsilon_{R_AO}}e^{-\varepsilon_D} \dfrac{1 + e^{-\varepsilon_L - \gamma}\left(1 + \dfrac{c}{K_A}\right)^2}{1 + e^{-\varepsilon_L}\left(1 + \dfrac{c}{K_A}\right)^2 + e^{-\varepsilon_D} + e^{-\varepsilon_D - \varepsilon_L - \gamma}\left(1 + \dfrac{c}{K_A}\right)^2}}. \tag{23}$$

*Equation 23* is the same as *Equation 16* only with $\frac{Re^{-\Delta\varepsilon_{RA}}}{N_{NS}}$ ($R^*$ of main text *Equation 1*) redefined as $[R_{tot}]e^{-\varepsilon_{R_AO}}$, and thus we'll arrive at the main text *Equation 1* following the same derivation steps leading from *Equations 16 to 13*.

Note that the difference in the interpretation of the $R^*$ term in main text *Equation 1* obtained from the derivations adopting the two perspectives demonstrated above doesn't affect our inference results based on the two-domain allosteric model (see the section '*Model parameter estimation*' below).

## Extended parametric study of main text Equation 1

### Effect of $\varepsilon_L$ on induction curve

As demonstrated in the main text section '*System-level ramifications of the two-domain model*' and main text *Figure 2A*, increasing $\varepsilon_L$ from the WT value alone can result in an induction curve featuring a sharply varying tail instead of the full sigmoidal shape (purple curve of main text *Figure 2A*). Further increasing $\varepsilon_L$, however, will result in a flat induction curve (orange curve of main text *Figure 2—figure supplement 3*), as the free energy gain of ligand binding becomes too limited to cause any noticeable effect on the allosteric response of the repressor even at its highest concentration used in our experiments ($c = 1000$ nM). Theoretically, based on main text *Equation 1*, an induction curve should always maintain the sigmoidal shape however high $\varepsilon_L$ becomes, given that we extend the concentration range for induction data measurement so that main text *Equation 2* is still applicable. Yet in practice, we find *E. coli* stops growing normally at $c$(aTC) >1000 nM, which is thus set as our concentration limit for induction data measurement.

### Effect of $\varepsilon_D$ on induction curve and the parametric degeneracy of flat induction curve

As demonstrated in the main text section '*System-level ramifications of the two-domain model*' and main text *Figure 2B*, decreasing $\varepsilon_D$ from the WT value alone simultaneously lowers the leakiness and the level of saturation of the induction curve while maintaining the sigmoidal shape (purple curve of main text *Figure 2B*). However, further decrease of $\varepsilon_D$ in principle can reduce their difference (dynamic range) to a level below the detection limit of our experiments as shown in *Equation 24*, and the induction curve will appear essentially flat (blue curve of main text *Figure 2—figure supplement 3*).

$$\lim_{c\to\infty} FC - FC(c = 0) = \frac{R^*(1 - e^{-\gamma})}{(1 + R^*e^{-\varepsilon_D - \gamma})(1 + R^*e^{-\varepsilon_D})} e^{-\varepsilon_D} \tag{24}$$

A flat induction curve can also result from vanishing inter-domain coupling ($\gamma$=0) where $FC$ becomes a constant $1/(1 + R^*e^{-\varepsilon_D})$ independent of $c$ (main text *Equation 1*), as shown by the green curve of main text *Figure 2—figure supplement 3*. Therefore, we've demonstrated the degeneracy of flat induction curves in the parameter space of the two-domain model, which prevent an accurate characterization of mutants with such induction curves (e.g. G102D, see main text *Figure 2—figure supplement 3*).

### $\gamma$ and the monotonicity of $FC(c)$

From main text *Equation 1*, we have

$$\frac{dFC(c)}{dc} = 2\frac{FC^2}{\left(1 + e^{-\varepsilon_L}\frac{c^2}{K^2}\right)^2} \frac{R^*e^{-\varepsilon_D - \varepsilon_L}c}{K^2}(1 - e^{-\gamma}) \tag{25}$$

As the fraction term of the RHS of *Equation 25* is positive besides at $c = 0$ (where it equals 0), the derivative of $FC$ with respect to $c$ is $\geq 0$, $\leq 0$, and =0 when $\gamma$>0, $\gamma$<0, and $\gamma$=0, respectively, for all ligand concentrations.

### Model parameter estimation

In this work, the estimation of allosteric parameters ($\varepsilon_D$, $\varepsilon_L$, and $\gamma$) of a mutant is divided into two steps: (1) calculation of $\varepsilon_D$ from the leakiness of its induction curve, which is then taken as a constant in step 2; (2) evaluation of $\varepsilon_L$ and $\gamma$ based on the full induction curve using the method of Bayesian inference. Detailed procedures are provided below.

### Calculation of $\varepsilon_D$

As shown in main text *Equation 1*, the leakiness of an induction curve is determined by $R^*$ and $\varepsilon_D$ of the corresponding mutant, where $R^*$, specified by *Equations 13 and 23*, is determined by system constants (e.g. $N_{NS}$ or cell volume), repressor copy number per cell (taken to be a constant across mutants; *Chure et al., 2019*), and the affinity between operator and the repressor with an active DBD. Therefore, since no mutation of direct DNA-interacting residues is made in the 24 mutants investigated in this work (*Ramos et al., 2005*), the variation of induction curve leakiness of a mutant

repressor from the WT is attributed solely to the change of $\varepsilon_D$. This enables the direct calculation of $\varepsilon_D$ of the mutants from their induction curve leakiness as given in **Equation 26**.

$$\epsilon_D = -ln\left(\frac{1}{FC(c=0)} - 1\right) + ln(R^*) \tag{26}$$

Although the exact value of $R^*$ is not resolved in our experiments, it doesn't affect the evaluation of the change of $\varepsilon_D$ in a mutant relative to the WT (main text **Figure 3** and **Figure 4—figure supplement 2**). This reflects how the mutations modify the allosteric nature of the repressor, which is what we focus on here (also see the next part of this section).

In main text **Figure 3** and **Figure 4—figure supplement 2**, the reported value of $\varepsilon_D$ is calculated based on the average of $FC(c=0)$ of four or more biological replicates for each mutant (except G102D-HQQ, C203V, and C203V-PIF, which have three replicates), with the reference $\varepsilon_D$(WT) set to 0. Error bar of $\varepsilon_D$ is calculated based on the standard error of the mean (SEM) of the corresponding leakiness measurement. The uncertainty of $\varepsilon_D$ is well below 0.1 $k_BT$ for all mutants except for C203V-PIF and Y132A-G102D-T26A, whose $\varepsilon_D$ uncertainties are 0.15 $k_BT$ and 0.2 $k_BT$, respectively. Thus, based on the characteristic effects of different allosteric parameters on the induction curve as established in main text **Equation 1**, the evaluation of $\varepsilon_D$ is decoupled from that of $\varepsilon_L$ and $\gamma$. This helps resolving the phenotype degeneracy in the parameter space of the two-domain model, enabling their estimation with high precision.

## Estimation of $\varepsilon_L$ and $\gamma$ with Bayesian inference

$\varepsilon_L$ and $\gamma$ of mutants are estimated simultaneously based on the induction curves following the standard workflow of Bayesian inference (see below) as introduced in previous works (**Chure et al., 2019**; **Schad et al., 2021**). The distinct effects of $\varepsilon_L$ and $\gamma$ on the induction curve (see main text section '*System-level ramifications of the two-domain model*') ensure their evaluation with a low level of uncertainty.

## Building a generative statistical model

In Bayesian inference, we want to estimate the value of $\varepsilon_L$ and $\gamma$ of a mutant given its induction data. To do this, we first need a statistical model that describes the conditional probability of different parameter values given the experimental observation (induction curve in this work). According to Bayes' theorem, such conditional probability (known as posterior distribution) is calculated as

$$p(\varepsilon_L, \gamma|y) = \frac{f(y|\varepsilon_L, \gamma)g(\varepsilon_L)g(\gamma)}{\iint d\varepsilon_L d\gamma \, f(y|\varepsilon_L, \gamma)g(\varepsilon_L)g(\gamma)} \tag{27}$$

Here, $y$ is the experimental data (fold change); $f(y|\varepsilon_L, \gamma)$ calculates the likelihood of observing $y$ given the value of $\varepsilon_L$, and $\gamma$ ; $g(\varepsilon_L)/g(\gamma)$ define the prior distribution of $\epsilon_L/\gamma$, which encodes our knowledge of the parameter value before seeing $y$. The denominator of the RHS of **Equation 27** only serves as a normalization factor and is treated as a constant. In practice, the proportionality relationship in **Equation 28** is sufficient for our purpose.

$$p(\varepsilon_L, \gamma|y) \propto f(y|\varepsilon_L, \gamma)g(\varepsilon_L)g(\gamma) \tag{28}$$

Next, we specify the likelihood function and the prior distributions at the RHS of **Equation 28**. Given the values of $\varepsilon_L$ and $\gamma$, we can calculate the expected fold change (μ) with main text **Equation 1**. In our experiments, however, several independent replicate measurements of fold change are made to suppress random error, which are expected to be normally distributed about the theoretical value μ. We thus have

$$f(y|\varepsilon_L, \gamma) = \frac{1}{(2\pi\sigma^2)^{N/2}} \prod_{i=1}^{N} exp\left(\frac{-[y_i - \mu(\varepsilon_L, \gamma)]^2}{2\sigma^2}\right) = Normal\left\{\mu(\varepsilon_L, \gamma), \sigma\right\}, \tag{29}$$

where $N$ is the total number of measurements made for $y$. As we've introduced an additional parameter $\sigma$ that describes the width of measurement distribution about the expected value in our statistical model, our complete posterior distribution becomes

$$p(\varepsilon_L, \gamma, \sigma|y) \propto f(y|\varepsilon_L, \gamma, \sigma)g(\varepsilon_L)g(\gamma)g(\sigma), \tag{30}$$

where $g(\sigma)$ is the prior distribution of $\sigma$.

Now with the likelihood function specified, our only task left before having a complete posterior distribution is to define the three prior distributions at the RHS of *Equation 30*.

As seen in main text *Equation 1*, $\varepsilon_L$ affects the gene expression level effectively through the composite factor $e^{-\varepsilon_L}/K^2$ in our two-domain model, which has to be evaluated as a whole in the Bayesian inference. Nonetheless, as only 3 of the total 24 mutants investigated in this work contain mutations of direct ligand-binding residues, the change of the inferred composite factor from the WT value for any mutant is attributed solely to the variation of $\varepsilon_L$ for intuitive comparison. However, we note that for the three mutants that do contain mutations of ligand-binding residues (P105M, I174K, and F177S) (*Leander et al., 2020*), such change could result from the variation of $K$ as well. For example, the triple mutant PIF has the highest $\varepsilon_L$ among all the queried mutants, which is intuitive considering that it contains mutations of two ligand-contacting residues.

In practice, we set $K$ to 1 nM for convenience and assign a Gaussian prior to $\varepsilon_L$ (*Equation 31*), which admits the ability of point mutations to change intra-domain energetics by a few $k_BT$ while permitting more extreme scenarios (*Chure et al., 2019*; *Daber et al., 2011*; *Reichheld et al., 2009*). The apparent binding affinity between our ligand (aTC) and TetR calculated with $K = 1$ nM and $\varepsilon_L$ of values within one standard deviation from the mean of g($\varepsilon_L$) well contain those of a range of TetR mutants reported in previous experimental works (*Scholz et al., 2000*; *Schubert et al., 2004*). Like in the case of $\varepsilon_D$, we focus on the difference between the inferred $\varepsilon_L$ of mutants and WT rather than their absolute value (main text *Figure 3* and *Figure 4—figure supplement 2*).

$$g(\varepsilon_L) = Normal\left\{5.5, 2.5\right\} \tag{31}$$

Similarly, we assign a normal distribution for the inter-domain coupling $\gamma$, which permits a cooperative free energy on the order of a few $k_BT$ (*Motlagh et al., 2014*), and the rare situation where a point mutation can reverse the sign of cooperativity between the two domains (*Scholz et al., 2004*).

$$g(\gamma) = Normal\left\{5, 2.5\right\} \tag{32}$$

Lastly, following *Chure et al., 2019*, the prior distribution of $\sigma$ is given by a half normal distribution (*Equation 33*), with $\varphi$=0.05. Such a choice restricts most fold change values generated by the statistical model to stay within the physical bounds of 0 and 1 (see main text *Equation 1*), while permitting rare exceptions that extend slightly beyond the bounds due to experimental noise.

$$g(\sigma) = \sqrt{\frac{2}{\pi\phi^2}}exp\left(\frac{-\sigma^2}{2\phi^2}\right) \tag{33}$$

To check the validity of the chosen prior distributions, we inspect if the simulated induction data based on them comply with our physical understanding of the system in the next step.

## Prior predictive checks

With the statistical model and the prior distributions in hand, we then simulate a set of induction data through such proposed data generation process, and check if the simulated fold changes stay mostly within the physical bounds of 0 and 1.

Specifically, we first draw 1000 sets of leakiness ($\varepsilon_D$), $\varepsilon_L$, $\gamma$, and $\sigma$ values from their prior distributions. For each set of parameters, we then calculate the expected fold change $\mu$ and draw four fold changes from the likelihood function (*Equation 29*) at each one of the 12 ligand concentrations used for experimental induction curve measurements. This matches the number of experimental measurements we made for most mutants. The prior distribution of leakiness is chosen as a half normal distribution centered at 0.005 with a standard deviation of 0.3 (see main text *Figure 3— figure supplement 2A*), which well covers the range of experimental leakiness we see and expect.

As shown in main text *Figure 3—figure supplement 2B*, the 5th percentile of the simulated fold change measurements has the characteristic shape of an induction curve. 95% of the simulated data falls between 0.05 and 1.1, and the 99th percentile extremes are bound between –0.1 and 1.2, which agree with our physical expectation considering the noise in biological measurements.

Satisfied with our prior choices, we go on to check the validity of the complete statistical model and our computational algorithm used for inference in the next step.

## Simulation-based calibration

With confidence in our chosen prior distributions, we proceed to check if our complete statistical model and computational algorithm allow for a faithful inference of the parameter values given the corresponding fold change data.

To do so, for each of the 1000 sets of parameters (leakiness, $\varepsilon_L$, $\gamma$, and $\sigma$) drawn for prior predictive check with its simulated data $\tilde{y}$, we estimate its posterior distribution p($\varepsilon_L$, $\gamma$, $\sigma|\tilde{y}$) given the leakiness and see how well we can recover the true parameter values. The posterior distributions are sampled using MCMC. Specifically, for each $\tilde{y}$, 1 million MC sweeps are performed to sample its posterior distribution. In each MC sweep, random moves of $\varepsilon_L$, $\gamma$, and $\sigma$ are proposed sequentially, which are accepted based on the probability specified in *Equation 30* following the Metropolis algorithm. Parameter values after every 1000 MC sweeps are recorded as the posterior samples (see the main text section '*Materials and methods*' for the specific code used for MC sampling).

To assess the validity of our complete statistical model and computational algorithm, the sampled posterior distributions for the 1000 prior predictive draws are then examined using several diagnostic methods.

First, for any Bayesian model, the posterior distribution averaged over prior predictive draws (of large enough sample size) should always recover the prior distribution, as is proven in *Equation 34*. Here, $\tilde{\theta}$ is the prior predictive draw of parameter, $\tilde{y}$ is its simulated data, and $\theta$ is the inferred parameter. Any deviation between the distribution of inferred $\theta$ and its prior distribution indicates mistakes in either the prior predictive sampling or the estimation of posterior distributions.

$$
\begin{aligned}
\pi(\theta) &= \iint d\tilde{y}d\tilde{\theta}\ p(\theta|\tilde{y})f(\tilde{y}|\tilde{\theta})g(\tilde{\theta}) \\
&= \int d\tilde{y}\frac{f(\tilde{y}|\theta)g(\theta)}{\pi(\tilde{y})}\pi(\tilde{y}) \\
&= g(\theta)
\end{aligned}
\tag{34}
$$

As shown in main text *Figure 3—figure supplement 3*, the average of our inferred posterior distributions (brown) of $\varepsilon_L$, $\gamma$, and $\sigma$ accurately recover the corresponding distributions of the ground truth values (blue) of the prior predictive draws. Therefore, our model well satisfies the self-consistency condition.

Second, we performed another ensemble level test of posterior distribution sampling using the rank statistics. Specifically, for each of the 1000 prior predictive draws of parameters $\tilde{\theta}$ ($\theta=\varepsilon_L$, $\gamma$, or $\sigma$) and its simulated data $\tilde{y}$, we've collected 1000 MC samples $\theta'_i$ ($i \in[1,1000]$) to estimate its posterior distribution $p(\theta'|\tilde{y})$ as detailed above. We then count how many of the posterior samples $\theta'_i$ are larger than the ground truth $\tilde{\theta}$, which is recorded as $r(\tilde{\theta})$ ($r(\tilde{\theta}) \in [0,1000]$). As proved by *Talts et al., 2018*, if $\theta'_i$ are sampled independently from the correct posterior distribution, the rank statistics ($r(\tilde{\theta})$) of the prior predictive draws should be uniformly distributed (over the integers $\{0, 1, ..., 1000\}$ in our case).

Several visualizations of the rank statistics of our prior predictive draws relative to the corresponding posterior samples are shown in main text *Figure 3—figure supplement 4*. Both the histograms (main text *Figure 3—figure supplement 4A*) and the empirical cumulative distribution function (ECDF) plots (main text *Figure 3—figure supplement 4B and C*) show that the rank statistics of $\varepsilon_L$, $\gamma$, and $\sigma$ are all uniformly distributed. While the histogram provides a general and interpretable way of checking uniformity, the ECDF is more sensitive to small deviations especially at small and large ranks. The green bands in main text *Figure 3—figure supplement 4A–C* show the 99th percentile expected from a true uniform distribution. The ECDF difference in main text *Figure 3—figure supplement 4C* is obtained by subtracting the theoretical cumulative distribution function of a uniform distribution from the observed ECDF, which makes deviations more evident if existent.

Third, besides examining the ensemble averaged behavior of the inferred posterior distributions, we further compute the posterior z-score and posterior contraction of each posterior distribution to see how well it recovers the true values of the corresponding parameters. As stated above, for each of the 1000 prior predictive draws of parameters $\tilde{\theta}$ ($\theta=\varepsilon_L$, $\gamma$, or $\sigma$) and its simulated data $\tilde{y}$, we've estimated its posterior distribution $p(\theta'|\tilde{y})$ using MCMC. The posterior z-score is defined as

$$z = \frac{M[p(\theta'|\tilde{y})] - \tilde{\theta}}{\sqrt{V[p(\theta'|\tilde{y})]}} \tag{35}$$

where $M$ and $V$ denote mean and variance, respectively. It quantifies how accurately the posterior recovers the true value of the inferred parameter. Apparently, a smaller/larger (absolute values of) z-score indicates that the posterior is concentrated around/away from the true parameter value. However, as posterior z-score reports the relative magnitude of the bias and width of the posterior distribution, it doesn't reflect the precision of the inference.

Another characterization of the posterior of a parameter, known as posterior contraction, is defined as the ratio of the posterior variance to the prior variance subtracted from one (*Equation 36*). Posterior contraction quantifies to what degree the data inform the posterior. Posterior contraction near 0 indicates that the posterior inference is largely influenced by the prior distributions, and is poorly informed by the data; while posterior contraction close to one indicates that the data are much more informative than the prior distributions.

$$c = 1 - \frac{V[p(\theta'|\tilde{y})]}{V[g(\theta)]} \tag{36}$$

The z-score and contraction analysis of posterior distributions can identify a series of pathologies of the statistical model (*Schad et al., 2021*). Ideally, a posterior distribution should stay close to [1,0] on the plane spanned by the axis of contraction and z-score, which means that the posterior accurately recovers the true parameter value with high precision.

The posterior z-score and contraction of 970 of our prior predictive draws are shown in main text *Figure 3—figure supplement 5A*, where z-scores of all parameters are clustered around 0, indicating high accuracy of their inferences. On the other hand, while the contractions of $\varepsilon_L$ and $\sigma$ are close to 1, the contraction of $\gamma$ is tailed with a minimum around 0.4 and a median of 0.972. Although the contraction of $\gamma$ is not as ideal as those of $\varepsilon_L$ and $\sigma$, it still reflects that the inference of $\gamma$ is reasonably well informed by the data (*Chure et al., 2019*; *Schad et al., 2021*).

In the above analysis, we excluded 30 prior predictive samples satisfying the criteria given in *Equation 37*, which essentially correspond to flat induction curve. The parameter values of these samples cannot be faithfully inferred due to the model degeneracy of flat induction curves as discussed in the section 'Extended parametric study of main text Equation 1' and main text . As the smallest *seperation* (*Equation 37*) of our experimental induction data is 8.6 (except that of G102D, see main text *Figure 2—figure supplement 3*), we don't encounter such problem in the inference using real data.

$$seperation = \left| \frac{\mu(c = 1000 \ nM) - \mu(c = 0)}{\sigma} \right| < 2 \tag{37}$$

It's noted that the contraction of $\gamma$, while reasonable, is not as ideal as those of other parameters (main text *Figure 3—figure supplement 5A*), which prompts a physical explanation. Accordingly, a closer examination of our physical model reveals the intrinsic difficulty of inferring $\gamma$ under two scenarios.

First, when the sum of $\gamma$ and $\varepsilon_L$ is large to the extent that the condition of *Equation 38* is satisfied, the main text *Equation 1* is reduced to *Equation 39*. For example, when the left-hand side of *Equation 38* is smaller than 0.05, the corresponding sum of $\varepsilon_L$ and $\gamma$ will be greater than 16.8 $k_BT$.

$$e^{-\varepsilon_L - \gamma} \left( \frac{c}{K} \right)^2 \ll 1, \ c = c^{max} = 1000 \ \text{nM} \tag{38}$$

$$FC = \left( 1 + \frac{R^* e^{-\varepsilon_D}}{1 + e^{-\varepsilon_L} \left( \frac{c}{K} \right)^2} \right)^{-1} \tag{39}$$

Under such scenario, the induction curve will be insensitive to the value of $\gamma$, as long as it's large enough so that *Equation 38* holds. This reversely causes the wider posterior distribution of $\gamma$ when it is inferred from such induction data, leading to its lower contraction compared with other parameters (*Equation 36*). On the other hand, *Equation 39* shows that its corresponding induction

curve will either saturate at $FC=1$, or has no saturation, featuring a sharply varying tail. We note that an induction curve with a sharply varying tail region may arise when **Equation 38** is not satisfied, under those conditions parameter inference does not suffer any difficulty.

Second, highly precise inference of $\gamma$ will also be difficult when **Equation 40** is true. Likewise, under such condition, main text **Equation 1** is effectively reduced to **Equation 39**, except that the corresponding induction curve is required to saturate at $FC=1$.

$$
\begin{cases}
e^{-\gamma} \ll 1, & e^{-\varepsilon_L}\left(\dfrac{c}{K}\right)^2 \gg R^* e^{-\varepsilon_D} \\
e^{-\varepsilon_L - \gamma}\left(\dfrac{c}{K}\right)^2 \ll 1, & \text{smaller c than required above.}
\end{cases}
\tag{40}
$$

Indeed, when we exclude the prior predictive draws with μ ($c$=1000 nM)>0.97 from the analysis, the average inferential performance of the model improved noticeably especially in the case of $\gamma$, whose new median contraction is above 0.997 (main text **Figure 3—figure supplement 5B**). The remaining few instances of lower $\gamma$ contraction still corresponds to large sums of $\gamma$ and $\varepsilon_L$, however, with the induction curve featuring a sharply varying tail region, as discussed above.

In our experimental dataset, such inference difficulty is only observed in the case of C203V, Y132A-C203V, and C203V-G102D-L146A due to their large $\gamma$ and $\gamma + \epsilon_L$ values (see main text **Figure 3**, **Figure 4**, and **Table 2**). As shown in main text **Table 2**, the inference results for the other 20 mutants stay highly precise and virtually unchanged after increasing the standard deviation of the Gaussian prior of $\gamma$ ($g_\gamma^{std}$) from 2.5 to 5 $k_B T$. This demonstrates that the inference results for these mutants are strongly informed by the induction data and there is no difficulty in the precise inference of the parameter values. On the other hand, the inferred $\gamma$ values (especially the upper bound of the 95% credible region) for C203V, Y132A-C203V, and C203V-G102D-L146A increased with $g_\gamma^{std}$. This is because the induction curves in these cases are not sensitive to the value of $\gamma$ given that it's large enough as discussed above. Hence, when unphysically large $\gamma$ values are permitted by the prior distribution, they could enter the posterior distribution as well.

Such difficulty in the precise inference of $\gamma$ values for these three mutants, however, doesn't compromise the ability of our model in accurately capturing the comprehensive set of induction data (see section 'Posterior predictive checks' below). Additionally, the increase of the inferred $\gamma$ value of C203V at the use of larger $g_\gamma^{std}$ complies with the results presented in main text **Figure 4**, which show that the effect of C203V on $\gamma$ tends to be compromised when combined with mutations closer to the domain interface.

Now with confidence in our statistical model and its computational implementation, as well as understanding of the inferential limitations intrinsic to the biophysical model, we proceed to see how well they capture the experimental data.

## Posterior predictive checks

We now apply our statistical model to the inference of the biophysical parameters of our experimentally tested mutants and inspect how well it captures the corresponding induction data. The same parameter estimation procedure is applied to all the 24 mutants tested except G102D (main text **Figure 3** and **Figure 4—figure supplement 2**), and we describe the case of G102D-Y42M-I57N here as an example.

First, the $\varepsilon_D$ of G102D-Y42M-I57N is calculated to be $1.63^{+0.01}_{-0.02}$ (main text **Figure 3**) based on its leakiness measurements of 0.0423±0.0005 (mean±SEM) and **Equation 26** (the $\varepsilon_D$ of WT is taken as 0 which has a leakiness of 0.0086). Next, with the average leakiness and the full induction data $y$, we generated 1000 sets of parameters $\{\varepsilon_L, \gamma, \sigma\}$ using MC sampling as estimation of the posterior distribution p$(\varepsilon_L, \gamma, \sigma|y)$. The reported inference result of an individual parameter $\theta$ is the median of the marginalized posterior $p(\theta|y)$ with the error bars showing the 95% credible interval (main text **Figure 3** and **Figure 4—figure supplement 2**).

To see how well the statistical model and the posterior parameters capture the experimental data from which they are inferred, we generated induction data using each of the 1000 posterior samples (like we did for the prior predictive draws) and inspect how well it recapitulates the experimental observation. The tight joint and marginal distributions in main text **Figure 3—figure supplement 6A** show that the parameters are inferred with high precision. A moderate correlation is observed between $\varepsilon_L$ and $\gamma$, while they are more symmetric to $\sigma$. More importantly, the simulated induction curves using the posterior samples follow the experimental measurements closely, which all fall within

the 99th percentile. Thus, our statistical model accurately describes the experimental observation, which is also shown in the posterior predictive checks of the other mutants (main text *Figure 3* and *Figure 4—figure supplement 2*).

## Prediction of mutation combinations based on the additive model

After characterization of the 15 mutants shown in main text *Figure 3*, we can generate prediction of combined mutants assuming additivity. Specifically, the biophysical parameters of a mutant containing two characterized mutations (mut1 and mut2) is predicted based on main text *Equations 4 and 5*. The two equations are first used to calculate the $\varepsilon_D$ of the combined mutant, and then applied to every one of the 1000×1000 pairs of the posterior samples ($\varepsilon_L$ and $\gamma$) of mut1 and mut2. We thus get the $\varepsilon_D$ value with 1 million sets of $\{\varepsilon_L, \gamma\}$ as estimation of the biophysical parameters of the combined mutant. The medians of the marginal distributions of $\varepsilon_L$ and $\gamma$ are then reported as the predicted parameter values for the combined mutant, where the error bars show the 95th percentile (main text *Figure 4—figure supplement 2*). In the basic additive model, we set $\alpha_{1,p} = \alpha_{2,p} = 1$ in main text *Equation 4*, where $p$ represents any one of $\varepsilon_D$, $\varepsilon_L$, and $\gamma$ (*Figure 4—figure supplements 1–2*). Systematic deviations between the experimental data of combined mutants and additive model prediction should manifest how important is epistasis between the mutations. This is rationalized in the modified additive model, where the values of $\alpha_{1,p}$ and $\alpha_{2,p}$ are adjusted to account for epistasis (main text *Figure 4*).

To assess the success of prediction, we compare the experimental induction curve with that generated using the parameters predicted by the (adjusted) additive model. Specifically, we calculate the theoretical fold change at the 12 ligand concentrations using the calculated $\varepsilon_D$ and each set of $\varepsilon_L$ and $\gamma$ with main text *Equation 1*. Here, $R^*$ is calculated to be 115 based on the WT leakiness, whose $\varepsilon_D$ is set to 0. The 95th percentile of thus generated 1 million induction curves are compared with the experimental induction data to assess the quality of prediction (main text *Figure 4* and *Figure 4—figure supplement 1*).

## Calculation of the apparent binding affinities to ligand and operator

The apparent biding affinity of TetR to the ligand (aTC) and operator (*tetO2*) used in our experiments are calculated as those of the $L_I D_I$ conformation of the repressor for it has the lowest free energy among all unbound states. Accordingly, the apparent dissociation constant of WT TetR to ligand is calculated as

$$K_{apparent}^{WT} = \sqrt{e^{\varepsilon_L^{WT}} \cdot 1} = 27.1 \text{ (nM)}, \tag{41}$$

which is converted to a binding free energy of –17.4 $k_B T$, close to the reported experimental value –16.8 $k_B T$ (*Schubert et al., 2004*).

Similarly, according to *Equation 23*, the apparent operator binding affinity of WT TetR is estimated as

$$\varepsilon_{R_A O} + \varepsilon_D = -ln\left( \frac{\frac{1}{leakiness} - 1}{[R_{tot}]} \right) \sim -16.4 \ (k_B T), \tag{42}$$

where $[R_{tot}]$ is evaluated based on the repressor copy number per cell of 5000 and the *E. coli* cell volume of 1 $\mu m^3$ (*Kubitschek and Friske, 1986*). This is comparable to the reported experimental value of –18.8 $k_B T$ (*Kedracka-Krok and Wasylewski, 1999*). However, it's noted that without direct measurement of the repressor concentration in the cell, this is only meant to be an order of magnitude comparison.

## Epistasis between C203V, Y132A, and other mutations

To probe the epistasis between mutations in TetR, we seek systematic deviations between the additive model prediction and experimental data of combined mutants. For example, our results indicate that quenching C203V's effect on $\varepsilon_D$ is a common modification of the additive model that improves its agreement with the experiments (main text *Figure 4* and *Figure 4—figure supplement 2*). The only exception is the mutant Y132A-C203V, where the additive model prediction on $\varepsilon_D$ is already close. However, we note that Y132A is the most distant mutation from the DNA-binding residues among all that are paired with C203V (main text *Figure 3—figure supplement 1*), residing

above the ligand as C203V does. Thus, it falls in line with the reasoning of the epistasis in other combined mutants containing C203V (main text *Figure 4*).

On the other hand, despite its proximity to the ligand-binding residues, we find that Y132A's effect on $\varepsilon_L$ tend to be compromised when combined with other mutants, as is strikingly reflected in Y132A-C203V (main text *Figure 4* and *Figure 4—figure supplement 2*). This is indeed a commonly observed trend in the epistasis of other Y132A containing combined mutants. Tuning down Y132A's contribution to $\varepsilon_L$ in all cases leads to better additive model prediction, while tuning down that of the other mutation (R49G and PIF) resulted in worse performance (main text *Figure 4* and *Figure 4—figure supplement 2*). Similar to the case of C203V (discussed in the main text), the common epistatic interactions observed for Y132A here offers a possible explanation for why it doesn't rescue a range of dead mutations despite being able to enhance the allosteric response of TetR by itself (*Leander et al., 2020*).

## Mutation selection for two-domain model analysis

In this work, there are 24 mutants studied in total including the WT, and they contain mutations at 21 WT residues. We did not perform model parameter inference for the mutant G102D because of its flat induction curve (see the section 'Effect of $\varepsilon_D$ on induction curve and the parametric degeneracy of flat induction curve' and main text *Figure 2—figure supplement 3*). Therefore, there are 23 mutants analyzed in main text *Figure 5*.

Measuring the induction curve of a mutant involves a significant amount of experimental effort, which therefore is hard to be extended to a large number of mutants. Nonetheless, we aim to compose a set of comprehensive induction data here for validating our two-domain model for TetR allostery. To this end, we picked 15 individual mutants in the first round of induction curve measurements, which contains mutations spanning different regions in the sequence and structure of TetR (main text *Figure 3—figure supplement 1*). Such broad distribution of mutations across LBD, DBD, and the domain interface could potentially lead to diverse induction curve shapes and mutant phenotypes for validating the two-domain model. Indeed, as discussed in the main text section 'Extensive induction curves fitting of TetR mutants', the diverse effects on induction curve from mutations perturbing different allosteric parameters predicted by the model are successfully observed in these 15 experimental induction curves. Additionally, 5 of the 15 mutants contain a dead-rescue mutation pair, which helps us validate the model prediction that a dead mutation could be rescued by rescuing mutations that perturb the allosteric parameters in various ways.

Eight mutation combinations were chosen for the second round of induction curve measurement for studying epistasis, where we paired up C203V and Y132A with mutations from different regions of the TetR structure. Such choice is largely based on two considerations. (1) As both C203V and Y132A greatly enhance the allosteric response of TetR, we want to probe why they cannot rescue a range of dead mutations as observed previously (*Leander et al., 2020*). (2) C203V and Y132A are the only two mutants that show enhanced allosteric response in the first round of analysis. Combining detrimental mutations of allostery in a combined mutant could potentially lead to near flat induction curve, which is less useful for inference (see the section 'Effect of $\varepsilon_D$ on induction curve and the parametric degeneracy of flat induction curve').

## The simplicity of the two-domain model

In this work, we aim to build a minimalist model for two-domain allostery with only the most essential parameters for capturing experimental data. Hence, the homodimeric nature of TetR is deliberately ignored here for the simplicity of the two-domain model, which helps promote its mechanistic clarity and potential transferability to other allosteric systems.

Moreover, fewer parameters are needed in a simpler model. Our two-domain model currently uses only three biophysical parameters, which are all demonstrated to have distinct influences on the induction curve (see the main text section 'System-level ramifications of the two-domain model'). This enables the inference of parameters with high precision for the mutants, and the quantification of the most essential mechanistic effects of their mutations, provided that the model is shown to accurately recapitulate the comprehensive dataset (main text *Figure 3* and *Figure 4—figure supplement 2*). Thus, we found it was unnecessary to add another parameter for explicitly describing inter-chain coupling, which would likely incur uncertainty in the inference of parameters due to the redundancy of their effects on induction data, and prevent the model from making faithful predictions.

From a more biological point of view, TetR is an obligate dimer, meaning that the two chains must synchronize for function, supporting the two-domain simplification of TetR for binding concerns. Additionally, as shown in the section 'Inclusion of single-ligand-bound state of repressor', incorporating the dimeric nature of TetR in our model by allowing partial ligand binding does not change the functional form of main text *Equation 1* in any practical sense.

In summary, we think that the value of a simple physical model is twofold (e.g. the paradigm Ising model in statistical physics and the classic MWC model), first, its mechanistic clarity and potential transferability makes it a useful conceptual framework for understanding complex systems and establishing universal rules by comparing seemingly unrelated phenomena; second, it provides useful insights and design principles of specific systems if it can quantitatively capture the corresponding experimental data. Thus, given the current experimental data set, we believe it is justified to keep the two-domain model in its current form, while additional experimental data could necessitate a more complex model for TetR allostery in the future.

