## [Editor Report · eLife assessment]

The study presents **valuable** findings where two-domain thermodynamic model for TetR accurately predicts in vivo phenotype changes brought about as a result of various mutations. The evidence provided is **compelling** and features the first innovative observations with a computational model that captures the structural behavior, much more than the current single-domain models.

---

## [Referee Report · Reviewer #1 (Public Review)]

Summary:

The authors' earlier deep mutational scanning work observed that allosteric mutations in TetR (the tetracycline repressor) and its homologous transcriptional factors are distributed across the structure instead of along the presumed allosteric pathways as commonly expected. Especially, in addition, the loss of the allosteric communications promoted by those mutations, was rescued by additional distributed mutations. Now the authors develop a two-domain thermodynamic model for TetR that explains these compelling data. The model is consistent with the in vivo phenotypes of the mutants with changes in parameters, which permits quantification. Taken together their work connects intra- and inter-domain allosteric regulation that correlate with structural features. This leads the authors to suggest broader applicability to other multidomain allosteric proteins.

Here the authors follow their first innovative observations with a computational model that captures the structural behavior, aiming to make it broadly applicable to multidomain proteins. Altogether, an innovative and potentially useful contribution.

Weaknesses:

None that I see, except that I hope that in the future, if possible, the authors would follow with additional proteins to further substantiate the model and show its broad applicability. I realize however the extensive work that this would entail.

---

## [Referee Report · Reviewer #2 (Public Review)]

Summary:

This combined experimental-theoretical paper introduces a novel two-domain statistical thermodynamic model (primarily Equation 1) to study allostery in generic systems but focusing here on the tetracycline repressor (TetR) family of transcription factors. This model, building on a function-centric approach, accurately captures induction data, maps mutants with precision, and reveals insights into epistasis between mutations.

Strengths:

The study contributes innovative modeling, successful data fitting, and valuable insights into the interconnectivity of allosteric networks, establishing a flexible and detailed framework for investigating TetR allostery. The manuscript is generally well-structured and communicates key findings effectively.

Comments on revised version:

I am happy with the changes made by the authors

---

## [Author Response]

The following is the authors’ response to the previous reviews.

We greatly appreciate the comments from the editor and the reviewers, based on which we have made the revisions. We have responded to all the questions and summarized the revisions below. The changes are also highlighted in the manuscript.

Additionally, we’ve noticed a few typos in the manuscript presented on the eLife website, which were not there in our originally submitted file.

(1) In both the “Full text” presented on the eLife website and the pdf file generated after clicking “Download”: the last FC1000 in the second paragraph of the “Extensive induction curves fitting of TetR mutants” section should be FC1000WT .(2) In the pdf file generated after clicking “Download”: the brackets are all incorrectly formatted in the captions of Figure 4 and Figure 3—figure supplement 6.
**eLife assessment**
The fundamental study presents a two-domain thermodynamic model for TetR which accurately predicts in vivo phenotype changes brought about as a result of various mutations. The evidence provided is solid and features the first innovative observations with a computational model that captures the structural behavior, much more than the current single-domain models.

We appreciate the supportive comments by the editor and reviewers.

**Public Reviews:**

**Reviewer #1 (Public Review):**
Summary:The authors’ earlier deep mutational scanning work observed that allosteric mutations in TetR (the tetracycline repressor) and its homologous transcriptional factors are distributed across the structure instead of along the presumed allosteric pathways as commonly expected. Especially, in addition, the loss of the allosteric communications promoted by those mutations, was rescued by additional distributed mutations. Now the authors develop a two-domain thermodynamic model for TetR that explains these compelling data. The model is consistent with the in vivo phenotypes of the mutants with changes in parameters, which permits quantification. Taken together their work connects intra- and inter-domain allosteric regulation that correlate with structural features. This leads the authors to suggest broader applicability to other multidomain allosteric proteins. Here the authors follow their first innovative observations with a computational model that captures the structural behavior, aiming to make it broadly applicable to multidomain proteins. Altogether, an innovative and potentially useful contribution.

We thank the reviewer for the supportive comments.

Weaknesses:None that I see, except that I hope that in the future, if possible, the authors would follow with additional proteins to further substantiate the model and show its broad applicability. I realize however the extensive work that this would entail.

We thank the reviewer for the supportive comments and the suggestion to extend the model to other proteins, which we indeed plan to pursue in future studies.

**Reviewer #2 (Public Review):**
Summary:This combined experimental-theoretical paper introduces a novel two-domain statistical thermodynamic model (primarily Equation 1) to study allostery in generic systems but focusing here on the tetracycline repressor (TetR) family of transcription factors. This model, building on a function-centric approach, accurately captures induction data, maps mutants with precision, and reveals insights into epistasis between mutations.Strengths:The study contributes innovative modeling, successful data fitting, and valuable insights into the interconnectivity of allosteric networks, establishing a flexible and detailed framework for investigating TetR allostery. The manuscript is generally well-structured and communicates key findings effectively.

We thank the reviewer for the supportive comments.

Weaknesses:The only minor weakness I found was that I still don’t have a better sense into (a) intuition and (b) mathematical derivation of Equation 1, which is so central to the work. I would recommend that the authors provide this early on in the main text.

We thank the reviewer for the suggestion. The full mathematical derivation of Equation 1 is given in the first section of the supplementary file. Given the length of the derivation, we think it’s better to keep it in the supplementary file rather than the main text. In the main text, the first subsection (overview of the two-domain thermodynamic model of allostery) of the Results section and the paragraph right before Equation 1 are meant for providing intuitive understandings of the two-domain model and the derivation of Equation 1, respectively.

We would also like to point the reviewer to Figure 2-figure supplement 2 and Equations (12) to (18) in the supplementary file for an alternative derivation. They show that the equilibria among all molecular species containing the operator are dictated by the binding free energies, the ligand concentration, and the allosteric parameters. The probability of an unbound operator (proportional to the probability that the promoter is bound by a RNA polymerase, or the gene expression level) can thus be calculated using Equation (12), which then leads to main text Equation 1 following the derivation given there.

Additionally, we’ve added a paragraph to the main text (line 248-260) to aid an intuitive understanding of Equation 1.

“The distinctive roles of the three biophysical parameter on the induction curve as stipulated in Equation 1 could be understood in an intuitive manner as well. First, the value of ε_D_ controls the intrinsic strength of binding of TetR to the operator, or the intrinsic difficulty for ligand to induce their separation. Therefore, it controls how tightly the downstream gene is regulated by TetR without ligands (reflected in leakiness) and affects the performance limit of ligands (reflected in saturation). Second, the value of ε_L_ controls how favorable ligand binding is in free energy. When ε_L_ increases, the binding of ligand at low concentrations become unfavorable, where the ligands cannot effectively bind to TetR to induce its separation from the operator. Therefore, the fold-change as a function of ligand concentration only starts to noticeably increase at higher ligand concentrations, resulting in larger EC_50_. Third, as discussed above, γ controls the level of anti-cooperativity between the ligand and operator binding of TetR, which is the basis of its allosteric regulation. In other words, γ controls how strongly ligand binding is incompatible with operator binding for TetR, hence it controls the performance limit of ligand (reflected in saturation).”

We hope that the reviewer will find this explanation helpful.

**Reviewer #3 (Public Review):**
Summary:Allosteric regulations are complicated in multi-domain proteins and many large-scale mutational data cannot be explained by current theoretical models, especially for those that are neither in the functional/allosteric sites nor on the allosteric pathways. This work provides a statistical thermodynamic model for a two-domain protein, in which one domain contains an effector binding site and the other domain contains a functional site. The authors build the model to explain the mutational experimental data of TetR, a transcriptional repress protein that contains a ligand and a DNA-binding domain. They incorporate three basic parameters, the energy change of the ligand and DNA binding domains before and after binding, and the coupling between the two domains to explain the free energy landscape of TetR’s conformational and binding states. They go further to quantitatively explain the in vivo expression level of the TetR-regulated gene by fitting into the induction curves of TetR mutants. The effects of most of the mutants studied could be well explained by the model. This approach can be extended to understand the allosteric regulation of other two-domain proteins, especially to explain the effects of widespread mutants not on the allosteric pathways. Strengths: The effects of mutations that are neither in the functional or allosteric sites nor in the allosteric pathways are difficult to explain and quantify. This work develops a statistical thermodynamic model to explain these complicated effects. For simple two-domain proteins, the model is quite clean and theoretically solid. For the real TetR protein that forms a dimeric structure containing two chains with each of them composed of two domains, the model can explain many of the experimental observations. The model separates intra and inter-domain influences that provide a novel angle to analyse allosteric effects in multi-domain proteins.

We thank the reviewer for the supportive comments.

Weaknesses:As mentioned above, the TetR protein is not a simple two-main protein, but forms a dimeric structure in which the DNA binding domain in each chain forms contacts with the ligand-binding domain in the other chain. In addition, the two ligand-binding domains have strong interactions. Without considering these interactions, especially those mutants that are on these interfaces, the model may be oversimplified for TetR.

We thank the reviewer for this valid concern and acknowledge that TetR is a homodimer. However, we’ve deliberately chosen to simplify this complexity in our model for the following reasons.

(1) In this work, we aim to build a minimalist model for two-domain allostery withonly the most essential parameters for capturing experimental data. The simplicity of the model helps promote its mechanistic clarity and potential transferability to other allosteric systems.

(2) Fewer parameters are needed in a simpler model. Our two-domain model currently uses only three biophysical parameters, which are all demonstrated to have distinct influences on the induction curve (see the main text section “System-level ramifications of the two-domain model”). This enables the inference of parameters with high precision for the mutants, and the quantification of the most essential mechanistic effects of their mutations, provided that the model is shown to accurately recapitulate the comprehensive dataset. Thus, we found it was unnecessary to add another parameter for explicitly describing inter-chain coupling, which would likely incur uncertainty in the inference of parameters due to the redundancy of their effects on induction data, and prevent the model from making faithful predictions.

(3) From a more biological point of view, TetR is an obligate dimer, meaning thatthe two chains must synchronize for function, supporting the two-domain simplification of TetR for binding concerns.

Additionally, as shown in the subsection “Inclusion of single-ligand-bound state of repressor” of section 1 of the supplementary file, incorporating the dimeric nature of TetR in our model by allowing partial ligand binding does not change the functional form of main text equation 1 in any practical sense. Therefore, considering all the factors stated above, we think that increasing the complexity of the two-domain model will only be necessary if additional data emerge to suggest the limitation of our model.

**Recommendations for the authors:**

**Reviewer #1 (Recommendations For The Authors):**
This is an excellent work. I have only one suggestion for the authors. Interestingly, the authors also note that the epistatic interactions that they obtain are consistent with the structural features of the protein, which is not surprising. Within this framework, have the authors considered rescue mutations? Please see for example PMID: 18195360 and PMID: 15683227. If I understand right, this might further extend the applicability of their model. If so, the authors may want to add a comment to that effect.

We thank the reviewer for the supportive comments and for pointing us to the useful references. We have added some comments to the main text regarding this point in line 332-336: “The diverse mechanistic origins of the rescuing mutations revealed here provide a rational basis for the broad distributions of such mutations. Integrating such thermodynamic analysis with structural and dynamic assessment of allosteric proteins for efficient and quantitative rescuing mutation design could present an interesting avenue for future research, particularly in the context of biomedical applications (PMID: 18195360, PMID: 15683227).”

**Reviewer #3 (Recommendations For The Authors):**
The authors should try to build a more realistic dimeric model for TetR to see if it could better explain experimental data. If it were too complicated for a revision, more discussions on the weakness of the current model should be given.

We thank the reviewer for this valid concern and for the suggestion. The reasons for refraining from increasing the complexity of the model are fully discussed in our response to the reviewer’s public review given above. Primarily, we think that the value of a simple physical model is two-fold (e.g., the paradigm Ising model in statistical physics and the classic MWC model), first, its mechanistic clarity and potential transferability makes it a useful conceptual framework for understanding complex systems and establishing universal rules by comparing seemingly unrelated phenomena; second, it provides useful insights and design principles of specific systems if it can quantitatively capture the corresponding experimental data. Thus, given the current experimental data set, we believe it is justified to keep the two-domain model in its current form, while additional experimental data could necessitate a more complex model for TetR allostery in the future. Relevant discussions are added to the main text (line 443-446) and section 8 of the supplementary file.

“It’s noted that the homodimeric nature of TetR is ignored in the current two-domain model to minimize the number of parameters, and additional experimental data could necessitate a more complex model for TetR allostery in the future (see supplementary file section 8 for more discussions).”

Minor issues:(1) There is an error in Figure 3A, the 13th and 14th subgraphs are the same and should be corrected.

We thank the reviewer for capturing this error, which has been corrected in the revised manuscript.

(2) The criteria for the selection of mutants for analysis should be clearly given. Apart from deleting mutants that are in direct contact with the ligand of DNA, how many mutants are left, and how far are they are from the two sites? In line 257, what are the criteria for selecting these 15 mutants? Similarly, in line 332, what are the criteria for selecting these 8 mutants?

We thank the reviewer for this comment. The data selection criteria are now added in section 7 of the supplementary file. The distances to the DNA operator and ligand of the 21 residues under mutational study are now added in Table 1 (Figure 3-figure supplement 9). The added materials are referenced in the main text where relevant.

“7. Mutation selection for two-domain model analysis

In this work, there are 24 mutants studied in total including the WT, and they contain mutations at 21 WT residues. We did not perform model parameter inference for the mutant G102D because of its flat induction curve (see the second subsection of section 2 and main text Figure 2—figure Supplement 3). Therefore, there are 23 mutants analyzed in main text Figure 5.

Measuring the induction curve of a mutant involves a significant amount of experimental effort, which therefore is hard to be extended to a large number of mutants. Nonetheless, we aim to compose a set of comprehensive induction data here for validating our two-domain model for TetR allostery. To this end, we picked 15 individual mutants in the first round of induction curve measurements, which contains mutations spanning different regions in the sequence and structure of TetR (main text Figure 3—figure Supplement 1). Such broad distribution of mutations across LBD, DBD and the domain interface could potentially lead to diverse induction curve shapes and mutant phenotypes for validating the two-domain model. Indeed, as discussed in the main text section "Extensive induction curves fitting of TetR mutants", the diverse effects on induction curve from mutations perturbing different allosteric parameters predicted by the model, are successfully observed in these 15 experimental induction curves. Additionally, 5 of the 15 mutants contain a dead-rescue mutation pair, which helps us validate the model prediction that a dead mutation could be rescued by rescuing mutations that perturb the allosteric parameters in various ways.

Eight mutation combinations were chosen for the second round of induction curve measurement for studying epistasis, where we paired up C203V and Y132A with mutations from different regions of the TetR structure. Such choice is largely based on two considerations. 1. As both C203V and Y132A greatly enhance the allosteric response of TetR, we want to probe why they cannot rescue a range of dead mutations as observed previously (PMID: 32999067). 2. C203V and Y132A are the only two mutants that show enhanced allosteric response in the first round of analysis. Combining detrimental mutations of allostery in a combined mutant could potentially lead to near flat induction curve, which is less useful for inference (see the second subsection of section 2).”

Since the number of hotspots identified by DMS is not very large, why not analyze them all?

We thank the reviewer for this comment. There are 41 hotspot residues in TetR(PMID: 36226916), which have 41*19=779 possible single mutations. It’s unfeasible to perform induction curve measurements for all of these 779 mutants in our current experiment. However, we agree that it would be helpful if we can obtain such a dataset in an efficient way.

In line 257, there are 15 mutants mentioned, while in Figure 5, there are 23 mutants mentioned, in Figure 3-figure supplement 1, there are 21 mutants mentioned, and in line 226 of the supplementary file, there are 24 mutants mentioned, which is very confusing. Therefore, the data selection criteria used in this article should be given.

We thank the reviewer for this comment. The data selection criteria are now given in section 7 of the supplementary file, which should clarify this confusion.

(3) In Figure 4 of the Exploring epistasis between mutations section, the 6 weights of the additive models corresponding to each mutation combination are different. On one hand, it seems that there are no universal laws in these experimental data. On the other hand, unique parameters of a single mutation combination were not validated in other mutation combinations, which somewhat weakened the conclusions about the potential physical significance of these additive weights.

We thank the reviewer for this comment. We admit that a quantitative universal law for tuning the 6 weights of the additive model does not manifest in our data, which indicates the mutation-specific nature of epistatic interactions in TetR as hinted in the different rescuing mutation distributions of different dead mutations (PMCID: PMC7568325). However, clear common trends in the weight tuning of combined mutants that contain common mutations do emerge, which comply with the structural features of the protein and provide explanations as to why C203V and Y132A don’t rescue a range of dead mutations (main text section “Exploring epistasis between mutations”). Additionally, the lack of a quantitative universal rule for tuning the 6 weights in our simple model doesn’t exclude the possibility of the existence of universal law for epistasis in TetR in another functional form, a point that could be explored in the future with more extensive joint experimental and computational investigations.

In Eq. (27) of the supplementary file, the prior distribution of inter-domain coupling γ is given as a Gaussian distribution centered at 5 kBT. Since the absolute value of γ is important, can the authors explain why the prior distribution of γ is set to this value and what happens if other values are used?

We thank the reviewer for the question. As explained in the corresponding discussions of Eq. (27) in the supplementary file, the prior of γ is chosen to serve as a soft constraint on its possible values based on the consideration that 1. inter-domain energetics for a TetR-like protein should be on the order of a few kBT; and 2. the prior distribution should reflect the experimental observation in the literature that γ has a small probability of adopting negative values upon mutations. Given our thorough validation of the statistical model and computational algorithm (see section 3 of the supplementary file), and the high precision in the parameter fitting results using experimental data (Figure 3 and Figure 4-figure supplement 2), we conclude that 1. the physical range of parameters encoded in their chosen prior distributions agrees well with the value reflected in the experimental data; 2. the inference results are predominantly informed by the data. Thus, changing the mean of the prior distribution of γ should not affect the inference results significantly given that it remains in the physical range.

This point is explicitly shown in the added Table 2 (Figure 3-figure supplement 10), where we compare the current Bayesian inference results with those obtained after increasing the standard deviation of the Gaussian prior of γ from 2.5 to 5 kBT. As shown in the table, most inference results stay virtually unchanged at the use of this less informative prior, which confirms that they are predominantly informed by the data. The only exceptions are the slight increase of the inferred γ values for C203V, C203V-Y132A and C203V-G102D-L146A, reflecting the intrinsic difficulty of precise inference of large γ values with our model, as is already discussed in the second subsection of section 3 of the supplementary file. However, such observations comply with the common trend of epistatic interactions involving C203V presented in the main text and don’t compromise the ability of our model to accurately capture the induction curves of mutants. Relevant discussions are now added to the second subsection of section 3 of the supplementary file (line 368-385).

“In our experimental dataset, such inference difficulty is only observed in the case of C203V, Y132A-C203V and C203V-G102D-L146A due to their large γ and γ + ε_L_ values (see main text Figure 3, Figure 3—figure Supplement 10 and Figure 4). As shown in main text Figure 3—figure Supplement 10, the inference results for the other 20 mutants stay highly precise and virtually unchanged after increasing the standard deviation of the Gaussian prior of γ (gstdγ ) from 2.5 to 5 k_B_T. This demonstrates that the inference results for these mutants are strongly informed by the induction data and there is no difficulty in the precise inference of the parameter values. On the other hand, the inferred γ values (especially the upper bound of the 95% credible region) for C203V, Y132A-C203V and C203V-G102D-L146A increased with gstdγ . This is because the induction curves in these cases are not sensitive to the value of γ given that it’s large enough as discussed above. Hence, when unphysically large γ values are permitted by the prior distribution, they could enter the posterior distribution as well. Such difficulty in the precise inference of γ values for these three mutants however, doesn’t compromise the ability of our model in accurately capturing the comprehensive set of induction data (see part iv below). Additionally, the increase of the inferred γ value of C203V at the use of larger gstdγ complies with the results presented in main text Figure 4, which show that the effect of C203V on γ tends to be compromised when combined with mutations closer to the domain interface."